

# SafeAuto: Knowledge-Enhanced Safe Autonomous Driving with Multimodal Foundation Models

**Jiawei Zhang** [1] [†] **Xuan Yang** [2] [‡] **Taiqi Wang** [3] [§] **Yu Yao** [3] [§] **Aleksandr Petiushko** [3] [§] **Bo Li** [1] [2] [4]

## Abstract

Traditional autonomous driving systems often struggle to connect high-level reasoning with low-level control, leading to suboptimal and sometimes unsafe behaviors. Recent advances in multimodal large language models (MLLMs), which process both visual and textual data, offer an opportunity to unify perception and reasoning. However, effectively embedding precise safety knowledge into MLLMs for autonomous driving remains a significant challenge. To address this, we propose SafeAuto, a framework that enhances MLLM-based autonomous driving by incorporating both unstructured and structured knowledge. First, we introduce a Position-Dependent Cross-Entropy (PDCE) loss to improve low-level control signal predictions when values are represented as text. Second, to explicitly integrate safety knowledge, we develop a reasoning component that translates traffic rules into first-order logic (e.g., "red light $\implies$ stop") and embeds them into a probabilistic graphical model (e.g., Markov Logic Network) to verify predicted actions using recognized environmental attributes. Additionally, our Multimodal Retrieval-Augmented Generation (RAG) model leverages video, control signals, and environmental attributes to learn from past driving experiences. Integrating PDCE, MLN, and Multimodal RAG, SafeAuto outperforms existing baselines across multiple datasets, enabling more accurate, reliable, and safer autonomous driving. The code is available at https://github.com/AI-secure/SafeAuto.

## 1. Introduction

Autonomous Driving (AD) systems (Kim et al., 2018; Jin et al., 2023; Hu et al., 2023; Jiang et al., 2023; Zhang et al., 2024b) have made significant strides in recent years, yet they often rely on separate modules for high-level decision-making (e.g., "the car should slow to a stop") and low-level control signal prediction (e.g., providing the specific speed or steering angle for the next few moments). However, these two aspects are inherently correlated, as high-level actions directly guide low-level control signals. This modular design often overlooks this correlation, leading to inefficiencies and less cohesive driving behaviors. Recent advancements in *Multimodal Large Language Models* (MLLMs) (Liu et al., 2023b;a; Lin et al., 2023) offer a promising avenue to bridge high-level reasoning and low-level control in autonomous driving (AD), which provide a unified framework capable of processing and reasoning over multiple data modalities, such as images, videos, and text. Some recent works (Wang et al., 2023; Xu et al., 2024; Wang et al., 2024) have begun to leverage MLLMs to generate both high-level action descriptions and low-level control signals in an end-to-end manner. However, they are predominantly data-driven and often fail to perform at human levels due to several limitations.

Firstly, for low-level action prediction, current approaches in adapting MLLMs generally follow two fashions. The first fashion treats the prediction of float numbers as text generation (Gruver et al., 2024; Xu et al., 2024), directly training the MLLM using cross-entropy (CE) loss for token prediction. Some variations (Brohan et al., 2023; Sima et al., 2023) of this method involve tokenizing the prediction range into several bins and adding new tokens for each bin into the LLM's vocabulary, allowing the model to predict the corresponding bin token ID. However, these methods remain somewhat coarse compared to traditional regression techniques (Hu et al., 2023) using Mean Squared Error (MSE) loss. Alternatively, another fashion (Jin et al., 2024b) employs a linear layer to decode the float number from the output hidden embeddings of the MLLM, enabling the use of MSE loss to train the model. While this approach may improve numerical accuracy, it compromises the autoregressive capability of the LLM, as the model can then only be purely used for numerical prediction and cannot

[†]Work done during an internship at Nuro,[‡]Work done during an internship at UIUC,[§]This work was done while the author was at Nuro [1]University of Chicago [2]University of Illinois Urbana-Champaign [3]Nuro [4]Virtue AI. Correspondence to: Jiawei Zhang <jwz@uchicago.edu>, Bo Li <bol@uchicago.edu>.

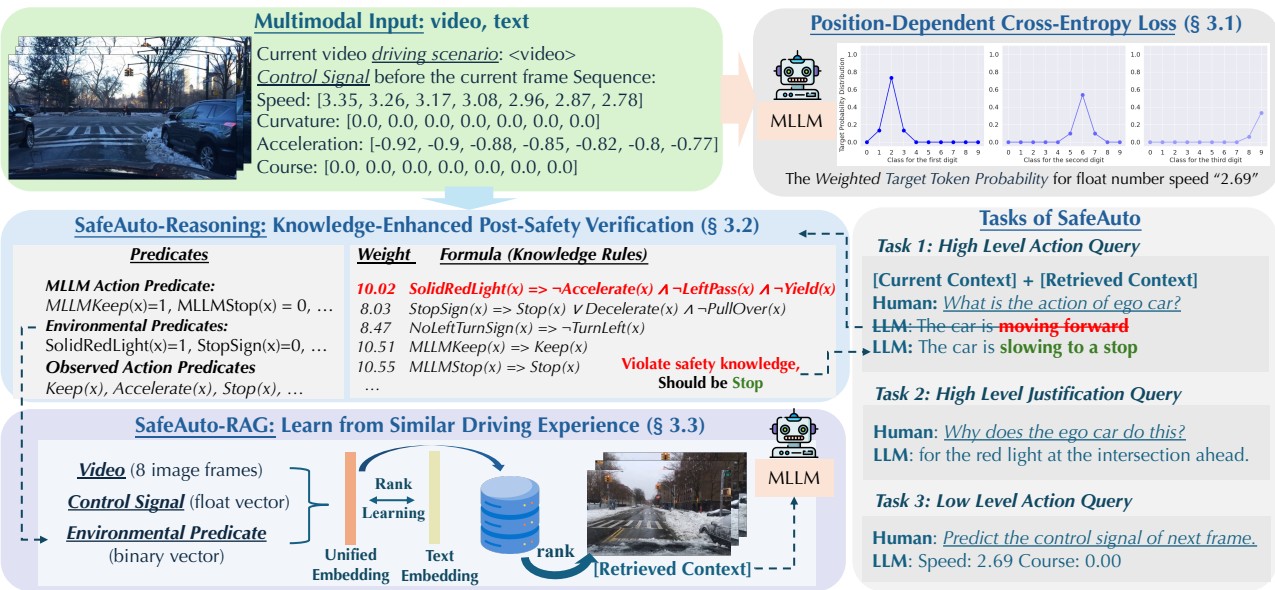

Figure 1: Overview of our SafeAuto pipeline for end-to-end high-level and low-level prediction in autonomous driving, featuring: (1) the Position-Dependent Cross-Entropy Loss (Section 3.1) for improved low-level numerical predictions using soft, weighted digit probability distributions; (2) Knowledge-Enhanced Post-Safety Verification (Section 3.2) with Markov Logic Networks to verify high-level actions against traffic rules; and (3) a Multimodal RAG (Section 3.3) training method that incorporates similar driving experiences via text-based rankings for better context-aware decision-making.

perform any further QA-for example, handling high-level question-answering. Additionally, regarding high-level action prediction, a significant limitation of current methods is their inability to effectively utilize both structured and unstructured knowledge when making decisions. Specifically, existing approaches often focus solely on data-driven techniques, inadequately incorporating structured knowledge such as traffic rules and safety constraints. Although some methods (Sima et al., 2023; Mao et al., 2023; Wang et al., 2024) attempt to include traffic regulations by embedding them into the model's context, this implicit approach is insufficient. Due to the inherent tendency of MLLMs to hallucinate, they may still generate unsafe or illegal actions. Meanwhile, while RAG (Lewis et al., 2020) has been employed in language models (Semnani et al., 2023; Zhang et al., 2024a) to mitigate issues like hallucination by incorporating relevant information from external sources, Yuan et al. (2024) first propose to combine the rich multimodal data inherent in autonomous driving contexts—such as videos, images, and control signals—to learn from past driving experiences as unstructured knowledge.

To address these challenges, we propose a novel framework SafeAuto that enhances MLLMs for autonomous driving through three key contributions as shown in Figure 1: (1) **Position-Dependent Cross-Entropy (PDCE) Loss:** We propose a PDCE loss that retains the autoregressive nature of the MLLM while behaving like an MSE loss during training. This loss function improves numerical prediction accuracy without compromising the model's language generation abil-

ities. (2) **Knowledge-Enhanced Post-Safety Verification:** We employ *Markov Logic Networks* (MLNs) (Richardson & Domingos, 2006) to explicitly encode domain knowledge and structured traffic rules into the decision-making process of the MLLM. This knowledge-enabled reasoning allows us to verify and correct the high-level actions suggested by the MLLM, ensuring they comply with traffic regulations and safety constraints. (3) **Multimodal RAG for Autonomous Driving:** We introduce a method that utilizes video data, control signals, and the environmental predicates used in the MLN to retrieve similar driving experiences. By learning a joint embedding across these modalities based on the ranking derived from text description of the current scenario—which contain rich semantic information—we can effectively leverage past experiences to inform current decision-making. By integrating these components, SafeAuto provides a comprehensive solution to the challenges faced by current MLLMs in autonomous driving. We evaluate our approach on two benchmark datasets: *BDD-X* (Kim et al., 2018) and *DriveLM* (Sima et al., 2023), both featuring low-level control signals and high-level action descriptions. Our experimental results demonstrate significant improvements in both low-level control accuracy and high-level action prediction. First, for low-level prediction on the BDD-X dataset, it reduces the Root Mean Square Error (RMSE) for speed and course predictions by an additional 5.8% and 14.1% over the state-of-the-art (SOTA) baselines, respectively. Furthermore, on the DriveLM dataset, it decreases the Average Displacement Error (ADE) for motion prediction by 44.4%. Second, for high-level prediction on

the BDD-X dataset, our method boosts high-level action performance beyond the SOTA by 28.0% under the CIDEr metric. Meanwhile, on the DriveLM dataset, it improves the high-level behavior prediction accuracy by an additional 13.0% over the SOTA.

## 2. Related Work

Advancements in autonomous driving have produced comprehensive frameworks like UniAD (Hu et al., 2023), which integrates modules for tracking, mapping, motion prediction, and occupancy estimation for low-level planning. However, UniAD lacks high-level action descriptions and textual justifications. To address high-level explanations, Kim et al. (2018) proposed an attention-based video-to-text model generating explanations of current driving actions. Similarly, ADAPT (Jin et al., 2023) employs a video Swin Transformer (Liu et al., 2022) to extract video tokens for separate high-level and low-level action predictions.

**Autonomous Driving with MLLM.** The emergence of MLLMs enables unified end-to-end generation of both high-level and low-level outputs. Most of these works often treat numerical control signals as text, training models using token prediction with cross-entropy loss. For example, DriveGPT4 (Xu et al., 2024) just treats low-level control signals as text, fine-tuning an MLLM to sequentially predict high-level and low-level actions in a conversational manner using the BDD-X dataset. DriveLM-Agent (Sima et al., 2023), influenced by RT-2 (Brohan et al., 2023), discretizes waypoints into bins, expanding the tokenizer vocabulary accordingly and fine-tuning the BLIP-2 (Li et al., 2023). While this facilitates end-to-end training, it remains coarse compared to UniAD (Hu et al., 2023), which uses MSE loss. Time-LLM (Jin et al., 2024b) decodes numerical predictions directly from output embeddings using a linear layer with MSE loss but diminishes the language model's autoregressive capabilities, limiting high-level question-answering abilities. Additionally, Tan et al. (2024) suggest that employing the LLM backbone in this way does not enhance regression performance. In contrast, we propose a novel PDCE loss that adapts the cross-entropy loss for numerical training to behave more like MSE loss while preserving the model's ability to perform high-level question-answering.

**Safety Guarantee.** Providing safety guarantees (Li et al., 2022; Zhang et al., 2023a) is always a fundamental concern in machine learning, especially in safety-critical applications like autonomous driving. Further advancements involve integrating perception and planning tools into the MLLM context. Agent-Driver (Mao et al., 2023) incorporates modules from UniAD into an MLLM framework, serving as a language agent for autonomous driving. OmniDrive (Wang et al., 2024) introduces a framework combining 3D perception, reasoning, and planning. However,

these methods remain purely data-driven and lack explicit safety verification for generated actions. Given the safety-critical nature of autonomous driving, ensuring that output actions are safe and compliant with traffic rules is essential. To address this, we incorporate extracted knowledge—specifically structured traffic rules—into a probabilistic graphical model like a Markov Logic Network (MLN) for explicit post-safety verification, which has been widely used in previous work (Yang et al., 2022; Zhang et al., 2023b). Besides, RAGDriver (Yuan et al., 2024) further enhances reasoning by retrieving similar driving experiences through triplet loss-based metric learning. We extend this approach by developing a more flexible and efficient retrieval system, directly training a joint embedding based on multimodal inputs to learn relative rankings from text similarity. Most importantly, we find that the incorporation of binary structured environmental predicates (e.g., the presence of a stop sign) from the previous reasoning components, namely MLNs, significantly improves retrieval performance.

## 3. SafeAuto

**Motivation.** Recent studies have begun to explore the integration of MLLMs into autonomous driving systems to enhance both high-level reasoning and low-level control actions. As illustrated in Figure 1, the MLLM receives a sequence of current driving images or videos, accompanied by textual descriptions of historical control signals, including speed, curvature, acceleration, and course, as inputs. Then, during the conversation, the model is expected to answer three types of queries: (1) _High-Level Action Queries_: These queries request a textual description of the action that the current ego vehicle is performing or should perform. For example, when asked *"What is the action of the ego car?"*, the MLLM is expected to respond with an answer like *"The car is slowing down to stop"*. (2) _High-Level Justification Queries_: These queries seek an explanation for the action provided by the MLLM. For instance, *"Why does the ego car do this?"* prompts the model to justify the action, such as *"for the red light at the intersection ahead."* (3) _Low-Level Action Queries_: These queries request specific control signals or trajectories that the vehicle should execute in the future. For example, the query *"Predict the control signals for the next frame"* would elicit a response like *"Speed: 2.69, Course: 0.00"*, which can then be translated into actual control commands for the autonomous vehicle. Typically, low-level action queries follow high-level action and justification queries, ensuring that the generated control signals are conditioned on prior high-level actions for more accurate and coherent driving control.

**Overview.** In this section, we detail the three main components proposed within this framework, each elaborated in subsequent sections: (1) a Position-Dependent Cross-Entropy Loss function for improved low-level action

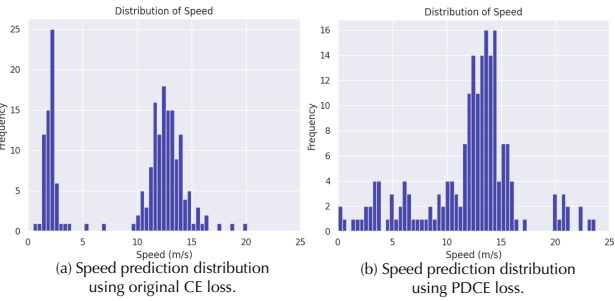

(a) Speed prediction distribution using original CE loss.

(b) Speed prediction distribution using PDCE loss.

Figure 2: Sampled speed prediction distribution under different losses when the ground truth is 12.46

prediction (Section 3.1); (2) Knowledge-Enhanced Post-Safety Verification using Markov Logic Network (MLN) for high-level action prediction (Section 3.2); (3) Multimodal Retrieval-Augmented Generation (RAG) for learning from similar driving experiences (Section 3.3). In summary, during training, we first fine-tune the MLLM using the PDCE loss with the retrieved context to enhance the accuracy of low-level action predictions. During evaluation, we retrieve the top $K$ similar driving experiences from the training database, generate high-level actions using the MLLM, and apply post-safety verification using the MLN to ensure that the actions comply with traffic rules and safety constraints.

### 3.1. SafeAuto—Position-Dependent CE loss

In existing approaches that utilize MLLMs for autonomous driving, the next-token prediction loss-specifically, the cross-entropy loss is commonly applied uniformly across all prediction tasks, including numerical value predictions. However, for numerical regression tasks, it is standard practice to use the Mean Squared Error (MSE) loss, as it directly penalizes the squared difference between the predicted and true values. A fundamental difference between CE loss and MSE loss lies in how they handle proximity to the target: MSE loss decreases as the prediction gets numerically closer to the target value, whereas CE loss does not necessarily exhibit this property. As a result, an issue is empirically observed in the speed prediction distribution when using the original CE loss to fine-tune the MLLM on the BDD-X dataset, as shown in Figure 2 (a), which displays predictions over 200 samples given the same input driving context with temperature as 1.0. As we can see, it reveals two distinct peaks, indicating that predictions closer to the ground truth value of "12.46" do not necessarily occur with higher frequency or lower loss, contrary to the behavior expected from MSE loss. A potential solution is to append an MLP to the MLLM to decode the output embeddings into float values, using MSE loss for fine-tuning. However, this will disrupt the MLLM's autoregressive token generation, transforming it into a pure transformer encoder (Tan et al., 2024) used only for regression tasks and losing its language generation capabilities necessary for high-level question-answering.

**PDCE loss.** To overcome these challenges, we adapt the CE loss to function more like MSE loss while maintaining textual predictions. Consider the previous example of predicting the float number "12.46." Originally, the MLLM is trained to maximize the probabilities $p('1')p('2' \mid '1')...p('6' \mid '12.4')$ by minimizing the CE loss with one-hot labels. However, this does not ensure that predictions closer to the target value have a lower loss as each digit's probability is treated with equal importance in loss.

To make the CE loss function behave more like MSE loss, we adhere to two key principles: (1) **Digit-Level Proximity**, where digits closer to the target in the same position incur lower loss, and (2) **Place-Level Importance**, where more significant digit positions have greater influence on the loss. To implement these, we introduce two modifications. First, *Digit-Level Loss Adjustment* replaces one-hot hard target labels with a soft target distribution $\mathcal{D}(\mu, \sigma)$ centered around the target digit $\mu$, e.g., a Gaussian distribution $\mathcal{G}(\mu, \sigma)$ to assign higher probabilities to numerically closer digits. The loss for each digit is then calculated as the Kullback-Leibler (KL) divergence between $\mathcal{D}(\mu, \sigma)$ and the predicted probability distribution $\mathcal{P}$ from the MLLM. Second, *Place-Level Weighting* assigns a weight $w_i$ to each digit position, aiming for weights to decrease with the position index $i$. Instead of using fixed weights, we adopt a soft approach by utilizing the approximated joint probabilities as weights, allowing flexible adjustment via $\sigma$. For example, in the number "12.46", the weight for digit '2' is the probability of '1' under $\mathcal{D}(1, \sigma)$, and the weight for digit '4' is the probability of '1' multiplied by the probability of '2' under $\mathcal{D}(2, \sigma)$. This ensures that later positions never have a higher influence than earlier ones and also allows for dynamic control of the decreasing weight with $\sigma$. Typically, as $\sigma$ approaches 0, the weighting scheme converges to the original CE loss. As a result, the final loss can be summarized as the weighted sum of the KL divergence between the probabilities generated by the MLLM and the target digit-level soft probability distributions. Mathematically, it can be expressed as:

$$\text{PDCE loss} = \sum_{i=1}^{n} w_i \cdot \text{KL}(\mathcal{P}_i \parallel \mathcal{D}(\mu_i, \sigma)), \qquad (1)$$

where $n$ is the number of digits used for representing the number, $\mu_i$ is the $i$-th digit, $\mathcal{P}_i$ represents the probability distribution over the possible digits for the $i$-th digit position from MLLM, $w_i = \prod_{j=1}^{i} \mathcal{D}(\mu_j, \sigma)[\mu_j]$. The pseudocode for implementing this loss during practice is provided in Appendix B. Besides, to balance the loss among various float numbers, we standardize their representation by using consistent digit lengths in text form. For example, in the BDD-X dataset, each number is formatted to five digits, such as representing 8.1 as "08.100" during training. The new prediction distribution using this loss with $\sigma = 0.35$ is demonstrated in Figure 2 (b). As shown, the distribution exhibits predictions that are more centered and closer to the

ground truth with a bell shape, which aligns with the desired outcome and verifies our intuition. Figure 7 demonstrates a further case study showing the token probability distribution when using CE loss and PDCE loss during training.

### 3.2. SafeAuto—reasoning

Currently, most methods for autonomous driving that utilize MLLMs are still purely data-driven. While these data-driven approaches have led to significant advancements, they may not be entirely suitable for safety-critical scenarios like autonomous driving, where reliability and strict adherence to safety regulations are paramount. To address this concern, we propose incorporating Probabilistic Graphical Models (PGMs) to verify the safety of the high-level actions suggested by the underlying MLLM. Specifically, in this paper, we focus on demonstrating how to adopt Markov Logic Networks to integrate domain knowledge and traffic rules into the decision-making process, while other variants are also applicable. In this section, we begin by explaining what MLNs are and how they apply to our AD context.

**Definition.** Essentially, an MLN consists of a set of first-order logic formulas, each associated with a weight that reflects the strength or confidence of that formula. These weights allow us to model uncertainty and handle exceptions in real-world knowledge. In our autonomous driving scenario, we use MLNs to model traffic rules and safety constraints. For example, a traffic rule like *"If there is a stop sign, then the vehicle should stop or decelerate"* can be represented as the logical formula: `StopSign(x)` $\implies$ `Stop(x)` $\vee$ `Decelerate(x)`, where x represents the current driving scenario. Here, predicates such as `StopSign(x)`, `Stop(x)`, and `Decelerate(x)` are logical functions that return true or false, indicating whether the condition holds in scenario x. Formally, in MLNs, *predicates* are logical functions defined over a set of constants $\mathcal{V} = \{v_1, v_2, \ldots, v_N\}$, where each $v_i$ represents an object or concept in the domain, such as "stop sign" or "red light." A predicate takes these constants as arguments and returns a truth value: $k(\cdot) : \mathcal{V} \times \cdots \times \mathcal{V} \to 0, 1$. While *formulas* are logical statements composed of predicates and logical connectives (e.g., $\implies$, $\wedge$, $\vee$), with each formula $f$ associated with a weight $w_f$ indicating its importance. Then, an MLN defines a joint probability distribution over all possible assignments of truth values to the ground predicates (predicates with specific constants assigned). The probability of a particular world (an assignment of truth values to all ground predicates) is given by: $P(\mathbf{X}) = \frac{1}{Z} \exp\left(\sum_{f \in \mathcal{F}} w_f \sum_{a_f \in \mathcal{A}_f} \phi_f(a_f)\right)$, where $\mathbf{X}$ is the set of all ground predicates, $\mathcal{F}$ is the set for all formulas $f$, $Z$ is the partition function ensuring the distribution sums to one, $\phi_f(a_f)$ is the potential function for formula $f$ with

assignment $a_f$ (which equals 1 if $f$ is true under $a_f$ and 0 otherwise), and $\mathcal{A}_f$ is the set of all possible assignments to the arguments of formula $f$.

**Autonomous Driving Context.** In our approach, we categorize predicates into *unobserved predicates* ($\mathcal{U}$), representing potential vehicle actions like `Accelerate(x)`, `Stop(x)`, and `TurnLeft(x)`, and *observed predicates* ($\mathcal{O}$). Observed predicates include (1) *MLLM Action Predicates*, such as `MLLMAccelerate(x)`, `MLLMStop(x)`, and `MLLMTurnLeft(x)`, which reflect high-level actions suggested by the MLLM. We map these to their truth values, introducing formulas like `MLLMAccelerate(x)` $\Rightarrow$ `Accelerate(x)` to align with MLLM's decisions. (2) *Environmental Predicates*, which describe conditions like `StopSign(x)` or `SolidRedLight(x)`, extracted from video data using object detectors. These predicates integrate with main action predicates to form logical formulas based on traffic rules from the California Driver Handbook [1], e.g., `StopSign(x)` $\implies$ `Stop(x)` $\vee$ `Decelerate(x)` $\wedge$ `¬PullOver(x)`. Additionally, predicates like `HCSTurnLeft(x)` reflect historical actions based on control signals, enhancing the vehicle's action decision-making process. Details are deferred to Appendix A.

**Inference.** Our goal is to infer the most probable assignment of the unobserved main action predicates $\mathcal{U}$ given the observed predicates $\mathcal{O}$. To determine the safest and most appropriate action, we perform inference by maximizing the conditional probability $P(\mathcal{U}|\mathcal{O})$. Specifically, we seek the assignment to the main action predicates $\mathcal{U}$ that maximizes this probability $\mathcal{U}^* = \arg\max_{\mathcal{U}} P(\mathcal{U}|\mathcal{O})$. Since the possible worlds for $\mathcal{U}$ (i.e., the possible assignments to the main action predicates) are inherently limited—a vehicle cannot simultaneously accelerate and decelerate or turn left and right—the inference process is thus computationally efficient. The detailed specifics of the possible worlds can be found in Appendix A.6.

**Training.** The training of the MLN is straightforward and involves learning the weights $w_f$ of the formulas to maximize $P(\mathcal{U}|\mathcal{O})$. In our approach, we utilize a mix of real and simulated data for training. The real data serves as the ground training data, provided by datasets such as BDD-X, while the simulated data allows us to model various driving conditions. This includes rare or dangerous scenarios not present in the real data, by simulating different truth values for the predicates to perform inference. Details are deferred to Appendix A.4.

**Safety Verification.** Initially, we collect observed grounded environmental predicates and the MLLM action predicates

---

[1] https://www.dmv.ca.gov/portal/handbook/california-driver-handbook/

from high-level actions generated by the MLLM, extracted through object detector and prompting with GPT4o. These predicates undergo inference within the trained MLN. If the MLN's final main action predicate output contradicts the MLLM's suggested action—suggesting a potential safety violation or a breach of critical traffic rules, we overwrite the original high-level action query based on the MLN's output and re-prompt the MLLM to generate a new high-level action, as depicted in Figure 1. Further details are available in Appendix A.5. In this way, the MLN serves as a post-verification layer that can override unsafe suggestions from the MLLM, enhancing the overall reliability of the autonomous driving system.

### 3.3. SafeAuto—Multimodal RAG

In this section, we introduce a novel training method for constructing a unified embedding that effectively integrates multiple modalities—current driving videos, historical control signals, and observed environmental predicate information from Section 3.2. Specifically, we aim to train the joint embedding to mirror the similarity rankings derived from the embedding of the textual descriptions for the current driving scenarios, which encapsulate the semantic information of all modalities during training. This approach facilitates the retrieval of similar driving experiences, enabling the ego vehicle to make more informed and context-aware decisions in current driving situations.

**Different Modality.** (1) *Image/ Video Embedding:* for the image or video modality, we utilize the pre-trained LanguageBind encoder (Zhu et al., 2024). This encoder processes an input image in $\mathbb{R}^{256 \times 1024}$, while processing video into eight frames and generates a video embedding in $\mathbb{R}^{2048 \times 1024}$. For simplicity and to reduce computational complexity, we apply global average pooling over the first dimension for both modalities here, resulting in a compressed embedding $\mathcal{Z}_v \in \mathbb{R}^{1 \times 1024}$ for use in subsequent experiments. (2) *Control Signal Vector:* the control signals are numerical values representing various aspects of the ego vehicle's historical state, such as speed, curvature, acceleration, and course. In datasets like BDD-X, each of these four types of control signals contains seven historical values (excluding the current frame), resulting in a total of $N = 4 \times 7 = 28$ values. We concatenate these values into a single vector $\mathcal{Z}_c \in \mathbb{R}^{1 \times N}$, which serves as the initial control signal vector. (3) *Environmental Predicate Vector:* These environmental predicates introduced in Section 3.2 are binary indicators of certain conditions or observations (e.g., presence of a stop sign, status of a traffic light). We encode these predicates into a single binary vector $\mathcal{Z}_p \in \{0,1\}^{1 \times M}$, where $M$ is the number of the whole environmental predicates. Empirically, we found that including this explicit binary representation significantly boosts retrieval performance, as demonstrated in Section 5. This enhancement

may be attributed to the reduction of noise inherent in the raw video embeddings or control signals; the binary predicates provide a clearer and more robust representation of essential environmental information.

**Unified Embedding Construction.** The central question is: *How can we train a unified embedding that effectively combines these different modalities for similarity computation and retrieval?* A key insight is that textual descriptions of the current driving scenario typically encompass all relevant semantic information, reflecting aspects of the video, control signals, and predicates. For instance, a text that concatenates action and justification—such as "*The car is slowing to a stop for the red light at the intersection ahead*" as shown in Figure 1 captures the essence of all three modalities. This comprehensive representation is particularly valuable for ranking the most similar driving scenarios. However, such ground text descriptions are often not available during evaluation. Building on this intuition, we propose learning a unified embedding that aligns these modalities in a shared space, akin to how text embeddings represent semantic information.

**Training Loss.** We start by mapping each input vector—$\mathcal{Z}_v$, $\mathcal{Z}_c$, and $\mathcal{Z}_p$—to aligned embeddings $\mathcal{Z}'_v$, $\mathcal{Z}'_c$, and $\mathcal{Z}'_p$ through individual projectors, each normalized to a unit $\ell_2$ norm and sharing the same dimension. We then apply weighting factors $w_v$, $w_c$, and $w_p$ to adjust the contributions of each modality in the final unified embedding, calculated as $\mathcal{Z}_u = \text{Projector}(w_v \mathcal{Z}'_v + w_c \mathcal{Z}'_c + w_p \mathcal{Z}'_p)$, which resides in $\mathbb{R}^{1 \times H}$. Motivated by CLIP (Radford et al., 2021), our objective is to train these projectors so that $\mathcal{Z}_u$ effectively mirrors the relational properties of text embeddings $\mathcal{Z}_t \in \mathbb{R}^{1 \times I}$ derived from high-level scenario descriptions, including actions and justifications. To achieve this, we randomly sample a batch of cases, $\mathcal{Z}'_u \in \mathbb{R}^{B \times H}$ and corresponding text embeddings $Z'_t \in \mathbb{R}^{B \times I}$ (assume each row has been normalized to a unit $\ell_2$ norm). We compute the similarity matrices $S'_u = \mathcal{Z}'_u (\mathcal{Z}'_u)^\top$ and $S'_t = Z'_t (Z'_t)^\top$. The training loss is then finally defined as the minimization of the divergence between the similarity matrix logits $S'_u$ and the temperature-scaled target logits $S'_t / \tau$, where $\tau$ is a temperature parameter that sharpens the focus on the most similar examples. This minimization ensures that the unified embeddings preserve the relative rankings of the text embeddings, crucial for effective retrieval tasks without ground textual descriptions during inference.

## 4. Experiments

In this section, we present our experimental results on two datasets: the BDD-X dataset (Kim et al., 2018) and the DriveLM dataset (Sima et al., 2023), both of which contain high-level action questions and low-level control questions. Specifically, we find that: (1) when using the Position-

Table 1: High-level action and justification evaluation on BDD-X dataset. B4, C, and M represent BLEU4, CIDEr, and METEOR, respectively.

| Method | Action | | | Justification | | |
|---|---|---|---|---|---|---|
| | B4 ↑ | C ↑ | M ↑ | B4 ↑ | C ↑ | M ↑ |
| ADAPT | 34.6 | 247.5 | 30.6 | **11.4** | 102.6 | **15.2** |
| DriveGPT4 | 30.0 | 214.0 | 29.8 | 9.4 | 102.7 | 14.6 |
| RAGDriver | 34.3 | 260.8 | 30.7 | 11.1 | **109.1** | 14.8 |
| SafeAuto | **38.6** | **337.4** | **35.5** | 9.4 | 96.0 | 14.0 |

Table 2: High-level behavior and low-level motion prediction evaluation on DriveLM dataset.

| Method | High-Level Behavior | | | Motion |
|---|---|---|---|---|
| | Acc ↑ | Speed ↑ | Steer ↑ | ADE ↓ |
| UniAD-Single | - | - | - | 1.80 |
| UniAD-Full | - | - | - | **0.80** |
| BLIP-RT-2 | - | - | - | 2.63 |
| DriveLM-Agent | 61.60 | 65.40 | 81.61 | 1.51 |
| SafeAuto | **74.60** | **81.61** | **81.90** | 0.84 |

Dependent Cross-Entropy loss, the numerical prediction of float numbers is significantly improved; (2) with the post-safety knowledge-enhanced verification via MLN, many dangerous high-level actions have been corrected; (3) the incorporation of Multimodal RAG, specifically integrating environmental predicate information from the MLN component, leads to significant improvements in the MLLM's high-level prediction performance. Notably, our framework is *plug-and-play* and can be directly applied to any new methods based on MLLMs. All experiments are conducted on eight NVIDIA A6000 GPUs.

**Datasets and Tasks.** (a) *BDD-X:* In this work, we adopt the processed version from RAGDriver (Yuan et al., 2024), where the task involves using an input video along with control signals from the past seven frames as context for a conversation that focuses on three types of questions: (i) high-level action queries, (ii) high-level justification queries, and (iii) low-level action predictions for speed and course in the next frame. This processed dataset contains 16,390 training video QA conversations and 2,123 test conversations. (b) *DriveLM:* The DriveLM dataset is built upon the nuScenes dataset (Caesar et al., 2020). In this work, we primarily focus on tasks that involve using six multi-view images from the current frame, and control signals including trajectory positions from the past three seconds as input context. The conversation concentrates on: (i) planning for possible high-level safe actions, (ii) high-level behavior involving predicting speed and steering actions, which serve as multiple-choice questions, and (iii) low-level motion, predicting 2D trajectories for the next three seconds, similar to UniAD (Hu et al., 2023). We filter instances to include only those with a prediction horizon of at least 3 seconds, resulting in a final dataset of 3,447 training conversations and 685 test conversations.

**Model.** We use the pretrained Video-LLaVA (Lin et al.,

2023) with Vicuna 1.5 7B (Zheng et al., 2023) as the base LLM for fine-tuning. We fine-tune the model for 2 epochs with a batch size of 128 on the BDD-X dataset and for 4 epochs with a batch size of 64 on the DriveLM dataset, using a learning rate of $5 \times 10^{-2}$.

**Experimental Details.** (a) *PDCE loss:* During the fine-tuning of the MLLM, we initialize $\sigma$ in $\mathcal{D}(\mu, \sigma)$ at a small value of 0.01 and geometrically increase it after each optimization step until it reaches the predefined value of $\sigma = 0.35$. This gradual increase helps stabilize the training process. (b) *Post-safety verification via MLN:* we fine-tune YOLOv8 (Jocher et al., 2023) using LISA dataset (Jensen et al., 2016) as the object detector for both traffic lights and signs. For the BDD-X dataset, we define 16 action predicates, 20 environmental predicates, and 35 formulas based on traffic rules. Similarly, for the DriveLM dataset, we define 7 action predicates, 29 environmental predicates, and 29 formulas. Further details are provided in Appendix A. (c) *Multimodal RAG:* we consistently employ four-layer multilayer perceptrons (MLPs) as projectors to obtain aligned embeddings for each modality and to generate the final unified embedding, and we use `sentence-t5-xl` (Ni et al., 2022) as our text encoder. The weighting factors $w_v$ and $w_c$ are both set to 0.4, while the weight for the predicate embedding $w_p$ is set to 0.2. We consistently set the learning rate to 0.001 and the temperature parameter $\tau$ to 0.5 for training. On the BDD-X dataset, the projectors are trained for 100 epochs with a batch size of 2,048; while for the DriveLM dataset, the projectors are also trained for 100 epochs but with a batch size of 512. Finally, we retrieve the Top $K = 2$ examples on BDD-X dataset, and Top $K = 1$ example for DriveLM dataset on finetuning MLLM and inference.

**Baselines.** (a) On the *BDD-X* dataset, we compare our method with several baselines: (1) *ADAPT* (Jin et al., 2023), a state-of-the-art video transformer-based method that provides high-level and low-level answers using two separate branches; (2) *TimeLLM* (Jin et al., 2024a), which repurposes frozen large language models for general time series forecasting by reprogramming numerical inputs into text-based patches; (3) *DriveGPT4* (Xu et al., 2024), the first work to provide both high-level action descriptions and low-level vehicle control signals in an end-to-end fashion using an MLLM; and (4) *RAGDriver* (Yuan et al., 2024), a state-of-the-art method that leverages triplet loss to train multimodal retrieval models for autonomous driving. (b) For the *DriveLM* dataset, we use: (1) *DriveLM-Agent*, the current state-of-the-art method that employs graph-based visual question answering to improve high-level responses and uses motion tokenization for low-level prediction; (2) *UniAD* (Hu et al., 2023), the state-of-the-art method on the nuScenes dataset used here for comparing low-level predictions——we consider two versions: UniAD (Full), which utilizes the entire historical video input, and UniAD (Single), a variant

Table 3: Low-level control signal prediction evaluation on BDD-X dataset.

| Method | Speed | | | | | | Course | | | | | |
|--------|-------|---|---|---|---|---|--------|---|---|---|---|---|
| | RMSE $\downarrow$ | $A_{0.1} \uparrow$ | $A_{0.5} \uparrow$ | $A_{1.0} \uparrow$ | $A_{5.0} \uparrow$ | $A_{10.0} \uparrow$ | RMSE $\downarrow$ | $A_{0.1} \uparrow$ | $A_{0.5} \uparrow$ | $A_{1.0} \uparrow$ | $A_{5.0} \uparrow$ | $A_{10.0} \uparrow$ |
| ADAPT | 2.68 | 11.77 | 31.79 | 47.48 | 92.75 | 95.87 | 5.87 | 54.49 | 86.39 | 91.06 | 97.36 | 98.20 |
| TimeLLM | 1.17 | 21.34 | 53.13 | 74.14 | 99.67 | 99.86 | 4.10 | 65.70 | 83.47 | 89.59 | 97.60 | 98.59 |
| DriveGPT4 | 1.09 | **56.93** | 77.77 | 87.97 | 99.00 | 99.57 | 4.57 | 69.22 | 79.14 | 84.47 | 95.72 | 96.74 |
| RAGDriver | 0.69 | 51.12 | 85.54 | 94.49 | **99.81** | **99.91** | 4.48 | 74.32 | 88.69 | 93.12 | **98.30** | 99.10 |
| SafeAuto | **0.65** | 55.49 | **88.84** | **95.34** | **99.81** | **99.91** | **3.85** | **76.26** | **89.68** | **94.11** | **98.30** | **99.25** |

Table 4: Ablation study of the contribution from each module in SafeAuto focusing on high-level action and justification assessment on the BDD-X dataset. "Acc" denotes the high-level action predicates accuracy.

| Method | Action | | | | Justification | | |
|--------|--------|---|---|------|---------------|---|---|
| | B4 $\uparrow$ | C $\uparrow$ | M $\uparrow$ | Acc $\uparrow$ | B4 $\uparrow$ | C $\uparrow$ | M $\uparrow$ |
| Base | 30.8 | 221.5 | 29.2 | 61.75 | 7.8 | 85.4 | 13.2 |
| PDCE | 31.4 | 231.4 | 29.3 | 61.94 | 7.9 | 84.2 | 13.2 |
| PDCE + MLN | 31.5 | 232.2 | 29.4 | 62.97 | 7.9 | 84.5 | 13.2 |
| PDCE + RAG | 38.2 | 334.8 | 35.3 | 91.00 | **9.4** | 95.5 | 13.9 |
| PDCE + MLN + RAG | **38.6** | **337.4** | **35.5** | **92.18** | **9.4** | **96.0** | **14.0** |

Table 5: Ablation study of the contribution from each module in SafeAuto on both high-level and low-level predictions using the DriveLM dataset.

| Method | High-Level Behavior | | | Motion |
|--------|---------------------|---|---|--------|
| | Acc $\uparrow$ | Speed $\uparrow$ | Steer $\uparrow$ | ADE $\downarrow$ |
| Base | 60.58 | 64.67 | 80.29 | 0.86 |
| PDCE | 63.21 | 67.88 | 79.27 | 0.85 |
| PDCE + MLN | 66.86 | 71.39 | 80.29 | 0.85 |
| PDCE + RAG | 74.01 | 79.27 | 81.61 | **0.84** |
| PDCE + MLN + RAG | **74.60** | **79.85** | **81.90** | **0.84** |

modified to use only the current frame's input for a fair comparison; and (3) *BLIP-RT-2*, which fine-tunes BLIP-2 (Li et al., 2023) on the DriveLM data and utilizes trajectory tokenization as proposed in RT-2 (Brohan et al., 2023).

**Metrics.** (a) For the *BDD-X* dataset, we adopt widely used metrics for high-level prediction, including 4-gram BLEU (B4) (Papineni et al., 2002), METEOR (M) (Banerjee & Lavie, 2005), and CIDEr (C) (Vedantam et al., 2015). For low-level prediction, we use the Root Mean Square Error (RMSE) for both steering angle (in degrees) and speed (in meters per second). We also present "tolerant accuracy" metrics, $A_\delta$, representing the accuracy of predictions when binarized as being within a tolerance threshold $\delta$ of the ground truth. (b) For the *DriveLM* dataset, the high-level behavior questions are multiple-choice problems concerning speed and steering. We report the overall accuracy, as well as individual accuracies for speed and steering predictions. For low-level trajectory prediction, we use the Average Displacement Error (ADE), as in UniAD, which indicates the average $\ell_2$ distance between the predicted trajectory and the ground truth trajectory and is calculated as the average of the errors at the 1st, 2nd, and 3rd seconds.

**Results.** (a) *BDD-X* Dataset: The final results for high-level prediction, including both action and justification, are presented in Table 1, while the low-level predictions for speed and course are shown in Table 3. For high-level action prediction, SafeAuto improves performance by 11.6%, 29.4%, and 15.6% for the BLEU4, CIDEr, and METEOR metrics, respectively. Although the justification performance is slightly lower than the state-of-the-art method, it still significantly outperforms the vanilla fine-tuned Video-LLaVA model, as demonstrated in Section 5. Additional case study is provided in Figure 5. For low-level control signal predic-

tion, SafeAuto achieves further reduction of 5.8% in RMSE for speed prediction and 14.1% in RMSE for course prediction. The contributions of each component to the overall performance are detailed in Section 5. (b) *DriveLM* Dataset: The final results are demonstrated in Table 2. For high-level behavior prediction, SafeAuto improves accuracy by 13.00% compared to the SOTA baseline DriveLM-Agent. For low-level motion prediction, it achieves a further reduction of 44.4% in ADE over the DriveLM-Agent. Notably, the ADE of SafeAuto is even comparable to UniAD (Full) which is trained purely for low-level prediction. Additional case study is provided in Figure 6.

## 5. Ablation Study

In this section, we conduct various ablation studies on SafeAuto to assess the impact of each module and different hyperparameters, as outlined in Section 3. For clarity, we refer to the base model—trained without any enhancements described in our paper—as 'Base'.

**Contribution from Each Module.** The contributions of different modules to both high-level and low-level performance on the DriveLM dataset are shown in Table 5, while the high-level results for the BDD-X dataset are presented in Table 4. The contributions of each module in SafeAuto to low-level control signal prediction on the BDD-X dataset are deferred to Appendix C.1. To accurately measure improvements in action performance, we introduce a new metric called *high-level action predicate accuracy* for the BDD-X dataset, which converts high-level action descriptions to one of 16 predefined actions using GPT4o strictly and measures accuracy. Our results reveal that: (1) PDCE loss markedly enhances low-level prediction while preserving high-level prediction performance; (2) post-safety verification via MLN

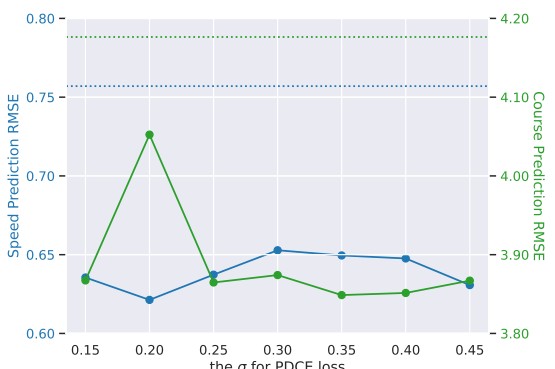

Figure 3: RMSE variation of low-level speed and course predictions with different PDCE loss $\sigma$ values on the BDD-X dataset. The dashed line represents the result of using the original CE loss.

Table 6: The impact of incorporating Environmental Predicates (EP) information for retrieval, along with the number of retrieved examples $K$ in Multimodal RAG, on high-level action and justification performance in the BDD-X dataset.

| Method | K | Action | | | | Justification | | |
|--------|---|--------|---|---|---|---------------|---|---|
| | | B4 ↑ | C ↑ | M ↑ | Acc ↑ | B4 ↑ | C ↑ | M ↑ |
| Base | - | 30.8 | 221.5 | 29.2 | 61.75 | 7.8 | 85.4 | 13.2 |
| RAG w/o EP | 1 | 29.4 | 219.2 | 28.5 | 59.06 | 7.3 | 74.8 | 12.6 |
| RAG w/o EP | 2 | 29.7 | 218.6 | 28.7 | 59.91 | 7.3 | 73.7 | 12.5 |
| RAG w/ EP | 1 | 38.1 | **334.8** | **35.4** | **91.47** | 8.8 | 89.2 | 13.5 |
| RAG w/ EP | 2 | **38.2** | **334.8** | 35.3 | 91.00 | **9.4** | **95.5** | **13.9** |

still corrects certain unsafe actions, even though the base model is conservative; the impact of each module on the rates of rule violation is detailed in Appendix C.3. (3) the use of Multimodal RAG significantly enhances performance, increasing high-level action predicate accuracy by 30% on BBD-X, and 14% on DriveLM.

**PDCE Loss with Different $\sigma$ Values.** We investigate the effect of varying $\sigma$ values on low-level predictions in the BDD-X dataset, as demonstrated in Figure 3. Our findings show that PDCE loss consistently achieves lower RMSEs for speed and course predictions than the CE loss, with minimal sensitivity to $\sigma$ changes, indicating strong stability.

**Influence of Environmental Predicates on Retrieval.** Our approach incorporates explicit Environmental Predicate (EP) information extracted from video and control signals. As indicated in Table 6, omitting environmental predicates yields performance akin to the base model, while including them markedly improves high-level prediction performance. This underscores the potential of using explicit binary environmental predicates to refine retrieval, eliminating the noise in the original data sources.

**Multimodal RAG with Different $K$.** We explore the impact of varying top $K$ selections for BDD-X dataset in Table 6. As we can see, significant improvements in high-level action prediction are achieved even with $K = 1$, and the performance is already comparable to the $K = 2$ scenario. Furthermore, selecting a larger $K$ value enhances performance in high-level justification prediction.

**Impact of Predicate Selection.** Predicates are crucial for retrieval and post-verification, as shown in an ablation study in Appendix C.2. This study highlights the importance of MLLM action and environmental predicates, demonstrating that certain environmental predicates significantly improve accuracy in driving scenarios when aligned with relevant traffic regulations for predicting lawful high-level actions.

**Case Study on Post-Safety Verification.** In the BDD-X dataset, the critical traffic rule `SolidRedLight(x)` $\implies$ `¬Accelerate(x) ∧ ¬LeftPass(x) ∧ ¬Yield(x)` and in the DriveLM dataset, the key rule is `RedYieldSign(x)` $\implies$ `¬Fast(x)`. These rules carry the highest weights in their respective MLNs. A case of MLN correcting aggressive driving behavior is illustrated in Figure 8.

## Limitation

There are some limitations in SafeAuto that could be addressed in future work. These include: (1) using better designed distribution $\mathcal{D}(\mu, \sigma)$ for PDCE loss to enhance performance, (2) improving the effectiveness of safety verification by refining predicate extraction, especially in scenarios with limited predicates, and (3) adding further filtering or reranking processes after retrieval in the Multimodal RAG within the MLLM context to enhance accuracy.

## Acknowledgment

This work is partially supported by the National Science Foundation under grant No. 1910100, No. 2046726, NSF AI Institute ACTION No. IIS-2229876, DARPA TIAMAT No. 80321, the National Aeronautics and Space Administration (NASA) under grant No. 80NSSC20M0229, ARL Grant W911NF-23-2-0137, Alfred P. Sloan Fellowship, the research grant from eBay, AI Safety Fund, Virtue AI, and Schmidt Science.

## Impact Statement

This paper presents work aimed at enhancing the safety and reliability of autonomous driving systems through the integration of multimodal foundation models, a newly designed loss function, structured safety knowledge based on traffic rules, and unstructured data from previous driving experiences. By improving the ability of autonomous vehicles to reason about complex driving scenarios, provide more accurate control signals, and adhere to traffic regulations, our framework, SafeAuto, has the potential to significantly reduce traffic accidents and enhance overall road safety with autonomous vehicles.

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

# A. Details on SafeAuto-Reasoning

## A.1. Traffic Rule Mapping

This section outlines the methodology for extracting first-order logic formulas from the California Driver Handbook [2]. Initially, all traffic rules are transformed into a structured format using GPT4o, based on the template: 'When [conditions], you should/should not [action] (unless [conditions]).' Subsequently, GPT4o is utilized again to translate the structured traffic rules into first-order logic formulas. The complete set of prompts is provided in Table 7 and Table 8.

```
As an agent for autonomous driving, your task is to extract pertinent rules from the provided text concerning autonomous driving, while
simultaneously filtering out irrelevant information. In specific, please extract rules from the text relating to specific driving
maneuvers listed as follows: keep, accelerate, decelerate, stop, make left turns, make right turns, reverse, merge, change lanes, park,
make U-turns, overtake, yield, follow different traffic signs. Disregard unrelated actions for autonomous driving like "looking around/
checking mirrors" or similar non-quantifiable action.

Use the structured format: 'When [conditions], you should/should not [action] (unless [conditions]).' Utilize 'OR' or 'AND' to connect
multiple conditions that may trigger the same action. Optionally, include 'unless [conditions]' where exceptions apply. Each rule
should be direct and applicable, ensuring it aids in the precise and safe execution of self-driving maneuvers. If the text does not
provide relevant advice for the actions listed, respond with 'None'.

Here is one example:

#Title#: Double Solid Yellow Lines
#Passage#: Do not pass over double solid yellow lines. Stay to the right of these lines unless you are:
 In a high-occupancy vehicle (HOV) carpool lane that has a designated entrance on the left.
 Instructed by construction or other signs to drive on the other side of the road because your side is closed or blocked.
 Turning left across a single set of double yellow lines to enter or exit a driveway or private road or make a U-turn.
Two sets of solid double yellow lines spaced two or more feet apart are considered a barrier. Do not drive on or over this barrier,
make a left turn, or make a U-turn across it, except at designated openings.
#Extracted Rules#: When driving near double solid yellow lines, you should stay to the right of these lines unless: (i) You are in a
high-occupancy vehicle (HOV) carpool lane that has a designated entrance on the left; (ii) You are instructed by construction or other
signs to drive on the other side of the road because your side is closed or blocked; (iii) You are turning left across a single set of
double yellow lines to enter or exit a driveway or private road, or to make a U-turn.
When two sets of solid double yellow lines spaced two or more feet apart are present, you should not drive on or over this barrier,
make a left turn, or make a U-turn across it, unless there is a designated opening for such maneuvers.

Now, extract the rules for the following passage:
#Title#: {title}
#Passage#: {passage}
#Extracted Rules#:
```

Table 7: Prompt for converting traffic rules to structured format

```
Your goal is to transform natural language driving rules into first-order logical rules for autonomous driving systems, start by
identifying the relevant actions and conditions from the text. Actions must choose from predefined predicates like Keep, Accelerate,
Decelerate, Stop, MakeLeftTurn, MakeRightTurn, Reverse, Merge, ChangeToLeftLane, ChangeToRightLane, Park, MakeUTurn, LeftPass,
RightPass and Yield.

First, analyze the natural driving rules to identify clear obligations (required actions) and prohibitions (banned actions), explicitly
ignoring any actions described as conditional permissions ("may"). Each rule will either dictate required actions under specific
conditions or explicitly ban certain actions in defined scenarios. For each rule:

Identify Required Actions (Obligations): If a rule specifies an action that must be taken under certain conditions, formulate this into
a logical statement using the format "Condition  Action." This represents an obligatory action.

Identify Prohibited Actions (Bans): If a rule bans certain actions in specific circumstances, express this as a logical statement using
the format "Condition  Action." This captures actions that are explicitly forbidden.

Here is one example:

#Natural Rules#: When driving near double solid yellow lines, you should stay to the right of these lines unless: (i) You are in a
high-occupancy vehicle (HOV) carpool lane that has a designated entrance on the left; (ii) You are instructed by construction or other
signs to drive on the other side of the road because your side is closed or blocked; (iii) You are turning left across a single set of
double yellow lines to enter or exit a driveway or private road, or to make a U-turn.
When two sets of solid double yellow lines spaced two or more feet apart are present, you should not drive on or over this barrier,
make a left turn, or make a U-turn across it, unless there is a designated opening for such maneuvers.
#Logical Rules#: (1) LeftSingleSetDoubleYellow  InHOVCarpoolWithLeftEntrance  Construction  ChangeToLeftLane  LeftPass
AdjacentSingleSetDoubleYellow  EnterOrExitDriveway  EnterOrExitPrivateRoad  MakeLeftTurn
(2) LeftDoubleSetsDoubleYellow  DesignatedOpeningLeftTurn  MakeLeftTurn
LeftDoubleSetsDoubleYellow  DesignatedOpeningUTurn  MakeUTurn

Now, extract the first-order logical rules for the following natural rules, and label each logical rule clearly with #Logical Rules#
and include an index that corresponds to the index of the original rule as shown in the example. Besides when there are only
conditioanl permissions ("may") and no clear obligations or progibitions, you can simply output None.
#Natural Rules#: {rules}
```

Table 8: Prompt for further converting traffic rules to first-order logic formulas

---

[2]https://www.dmv.ca.gov/portal/handbook/california-driver-handbook/

## A.2. YOLOv8 Fine-tuning

We fine-tuned the YOLOv8 model using the LISA dataset (Jensen et al., 2016), which contains annotations for both traffic signs and traffic signals. The dataset comprises four daytime sequences and two nighttime sequences, primarily designated for testing purposes, with a total duration of 23 minutes and 25 seconds of driving footage recorded in Pacific Beach and La Jolla, San Diego. It contains $43,007$ frames, annotated with $113,888$ traffic lights and $7,855$ traffic signs across $6,610$ frames. The YOLOv8m model was fine-tuned over $500$ epochs, with an input image resolution of $640 \times 640$ pixels.

## A.3. Predicate Extraction

For environmental predicates, we use the fine-tuned YOLOv8 for detection, as described in Appendix A.2. To ensure consistency with RagDriver (Yuan et al., 2024), we uniformly divide video segments into 8 frames and select the final frame as input. Additionally, in DriveLM, images from three perspectivesthe front camera, left front camera, and right front cameraare utilized. We leveraged the nuScenes map expansion to extract lane line information for both sides of the lane in which the ego vehicle is positioned. Historical control signals in BDD-X and DriveLM were obtained by querying GPT4o and mapping the results to corresponding environmental predicates (e.g., `HCSKeep(x)`). Specific details of the prompts used in this extraction process are provided in Table 9 and Table 10.

```
Given the current speed, curvature, acceleration, and course of the car, use one velocity predicate and one directional predicate to
best describe the behavior of the car.
The velocity predicates are: Keep, Accelerate, Decelerate, Stop, Reverse.
The directional predicates are: Straight, Left, Right.
Output the predicates directly without any additional information.
Here are some examples:
#Speed#: [7.18, 5.76, 4.45, 3.30, 2.24, 1.20, 0.36]
#Curvature#: [1.32, 0.88, 0.58, 1.85, 2.74, 1.61, 0.64]
#Acceleration#: [-1.22, -1.85, -2.39, -2.22, -2.01, -1.46, -0.87]
#Course#: [0.00, -10.03, -8.33, -3.23, -0.97, -0.32, -0.08]
#Predicate#: HCSStop, HCSLeft
#Speed#: [12.31, 9.51, 7.24, 5.38, 3.67, 2.76, 3.00]
#Curvature#: [-0.00, 0.00, 0.00, -0.05, -0.18, -0.67, -0.79]
#Acceleration#: [-1.85, -2.79, -2.73, -2.23, -1.67, -0.47, 0.71]
#Course#: [0.00, 0.00, 0.00, 0.00, -20.26, -60.78, 7.17]
#Predicate#: HCSDecelerate, HCSRight
#Speed#: [1.27, 4.18, 6.83, 8.87, 10.44, 12.22, 14.45]
#Curvature#: [0.00, 0.00, 0.00, -0.00, -0.01, -0.00, -0.00]
#Acceleration#: [2.27, 2.15, 1.81, 1.35, 1.28, 1.56, 1.45]
#Course#: [0.00, -0.09, 0.00, 0.00, 0.20, 0.00, 0.00]
#Predicate#: HCSAccelerate, HCSStraight
#Speed#: {speed}
#Curvature#: {curvature}
#Acceleration#: {acceleration}
#Course#: {course}
#Predicate:
```

Table 9: Prompt for Extracting High-level Control Signal Environmental Predicates from the BDD-X Dataset

```
Given the current speed and course of the car, use one velocity predicate and one directional predicate to best describe the behavior
of the car.
The velocity predicates are: Normal, Fast, Slow, Stop.
The directional predicates are: Straight, Left, Right.
Output the predicates directly without any additional information.
Here are some examples:
#Speed#: [(4.54, 0.0), (5.34, 0.0), (5.67, 0.0), (5.7, 0.0), (6.46, 0.0), (6.63, 0.0)]
#Course#: [(1.0, 0.0), (1.0, 0.0), (1.0, 0.0), (1.0, 0.0), (1.0, 0.0), (1.0, 0.0)]
#Predicate#: HCSFast, HCSStraight
#Speed#: [(10.01, 0.0), (9.88, 0.0), (9.52, 0.0), (9.39, 0.0), (9.15, 0.0), (8.94, 0.0)]
#Course#: [(0.84, 0.0), (0.84, 0.0), (0.86, 0.0), (0.89, 0.0), (0.93, 0.0), (0.95, 0.0)]
#Predicate#: HCSFast, HCSRight
#Speed#: [(2.51, 0.0), (2.49, 0.0), (2.45, 0.0), (2.43, 0.0), (2.43, 0.0), (2.37, 0.0)]
#Course#: [(0.85, 0.0), (0.85, 0.0), (0.86, 0.0), (0.85, 0.0), (0.82, 0.0), (0.75, 0.0)]
#Predicate#: HCSSlowly, HCSLeft
#Speed#: [(1.65, 0.0), (1.37, 0.0), (0.73, 0.0), (0.09, 0.0), (0.0, 0.0), (0.0, 0.0), (0.0, 0.0), (0.0, 0.0)]
#Course#: [(0.86, 0.0), (0.86, 0.0), (0.87, 0.0), (0.86, 0.0), (0.86, 0.0), (0.86, 0.0), (0.85, 0.0), (0.84, 0.0)]
#Predicate#: HCSStop, HCSStraight
#Speed#: {speed}
#Course#: {course}
#Predicate#:
```

Table 10: Prompt for Extracting High-level Control Signal Environmental Predicates from the DriveLM Dataset

With respect to MLLM action predicates, since the output of MLLM consists of high-level action descriptions such as "The car is slowing down to stop, we map these to predicates represented as (`MLLMDecelerate(x)`, `MLLMStop(x)`).

```
Given the current behavior of the car, please use predicates below to best describe the behavior of the car. The predicates are:
Keep, Accelerate, Decelerate, Stop, Reverse, TurnLeft, TurnRight, UTurn, Merge, LeftPass, RightPass, Yield, ChangeToLeftLane,
ChangeToRightLane, Park, PullOver.
Here are some examples:
#Current Behavior#: The car is travelling down the road.
#Predicates#: Keep
#Current Behavior#: The car is making left turn.
#Predicates#: TurnLeft
#Current Behavior#: The car is slowing down and then comes to a stop.
#Predicates#: Decelerate, Stop
#Current Behavior#: The car is accelerating and then turns right.
#Predicates#: Accelerate, TurnRight
#Current Behavior#: The car is making a left turn and accelerates.
#Predicates#: TurnLeft, Accelerate
#Current Behavior#: The car decelerates and stops.
#Predicates#: Decelerate, Stop

Now the current behavior of the car is described, provide the predicates that best describe the behavior of the car.

#Current Behavior#: {action}
#Predicates#:
```

Table 11: Prompt for Extracting Environmental Predicates from the BDD-X Dataset

In the BDD-X dataset, due to the increased number and complexity of high-level action descriptions for MLLM action predicates, we employ GPT4o with specifically designed prompts to extract these predicates, with the detailed prompts provided in Table 11. In DriveLM, given that the question-and-answer format comprises multiple-choice questions with fixed option descriptions, we predefine mapping rules to translate high-level action descriptions into predicates, as described in Table 12.

Table 12: Mapping of High-level Action Descriptions to MLLM Action Predicates.

| High-level Action Description | MLLM Action Predicate |
| --- | --- |
| Going straight
Slightly steering to the left
Slightly steering to the right | `MLLMStraight(x)` |
| Driving fast
Driving very fast | `MLLMFast(x)` |
| Driving slowly | `MLLMSlow(x)` |
| Driving with normal speed | `MLLMNormal(x)` |
| Not moving | `MLLMStop(x)` |
| Steering to the left | `MLLMLeft(x)` |
| Steering to the right | `MLLMRight(x)` |

### A.4. Training Details

The learning rate for the Markov Logic Network (MLN) is set at $1 \times 10^{-5}$. To mitigate the risk of overfitting and to avoid excessive reliance on frequently occurring scenarios, such as straight movements, regularization is incorporated into the training process, also set at $1 \times 10^{-5}$. The models are trained for a total of 300 epochs, unless interrupted by a predefined early stopping criterion: specifically, if the model's accuracy fails to improve by more than $1 \times 10^{-6}$ over 10 consecutive epochs, training will be terminated.

### A.5. Post-verification Details

As outlined in Section 3.2, during safety verification, we initiate the process by extracting observed grounded environmental predicates and MLLM action predicates using the object detector and GPT4o. If the final main action predicate output of the Markov Logic Network (MLN) conflicts with the suggested action from MLLM, we modify the high-level action query based on the output of the MLN. In the BDD-X dataset, we replace the original high-level action queries with new actions inferred from the MLN. For example, if the MLN predicts the possible world represented as "Stop(x) = 1" with the highest probability, we append the suggestion *"The ego vehicle should stop"* to the high-level action query. This approach

```
# str_num: a string representing a float number (excluding
the decimal point '.') with N digits
# logits: the logits distribution output from MLLM for each
digit in str_num, with a shape of N * 10
# sigma: the standard deviation of the Gaussian distribution

# Precompute the digit-level probability distributions
from scipy.stats import norm
distribution_dict = {}
for num in range(10):
  prob_distribution = np.array([norm(num, sigma).cdf(i + 0.5)
    - norm(num, sigma).cdf(i - 0.5) for i in range(10)])
  prob_distribution /= prob_distribution.sum()
  distribution_dict[str(num)] = prob_distribution

# Calculate weights for each digit position
tgt_probs = []
Weight = 1.0
for digit in str_num:
  # The place-level weighting
  digit_probs = distribution_dict[digit] * weight
  weight *= digit_probs[int(digit)]
  tgt_probs.append(digit_probs)

tgt_probs = np.array(tgt_probs)
# Compute the KL loss, constants are ignored
loss = - (tgt_probs * log_softmax(logits, axis=1)).sum()
```

Figure 4: The numpy-style pseudocode on PDCE loss.

facilitates the mapping back to the corresponding high-level action description and ensures the flow of conversation for subsequent queries.

In DriveLM, since high-level action queries are presented in a multiple-choice format, the final main action predicate output from the Markov Logic Network (MLN) may not always align directly with one of the options. In such cases, we filter the available options by the probability of possible worlds. Given that MLLM action predicates may map to multiple high-level action descriptions, it is feasible for multiple valid options to arise simultaneously. We then overwrite the high-level action queries by removing incorrect options and prompt the MLLM to regenerate an option.

### A.6. Predicates and Traffic Rules

This section provides a detailed overview of the specific aspects of the MLN construction for both the BDD-X and DriveLM datasets. Table 16, Table 17, Table 18, present the predicate set of BDD-X, all possible worlds, and first-order predicate logic, respectively; while Table 19, Table 20, and Table 21 show those of DriveLM.

## B. Pseudo-Code of PDCE loss

The pseudo-code for calculating the PDCE loss is provided in Figure 4.

## C. Experiements on Ablation Study

### C.1. Contribution from different module

Table 13 presents an ablation study evaluating the contribution of each module in SafeAuto to the low-level control signal prediction on the BDD-X dataset. Interestingly, we find that the MLN reasoning and RAG modules have only a minimal impact on the low-level prediction accuracy, with the primary improvement stemming from the PDCE loss, as expected. Additionally, we observe that incorporating RAG slightly increases the RMSE for speed prediction but decreases the RMSE for course prediction.

Table 13: Ablation study of the contribution from each module in SafeAuto focusing on low-level control signal assessment on the BDD-X dataset.

| Method | Speed | | | | | | Course | | | | | |
|---|---|---|---|---|---|---|---|---|---|---|---|---|
| | RMSE $\downarrow$ | $A_{0.1} \uparrow$ | $A_{0.5} \uparrow$ | $A_{1.0} \uparrow$ | $A_{5.0} \uparrow$ | $A_{10.0} \uparrow$ | RMSE $\downarrow$ | $A_{0.1} \uparrow$ | $A_{0.5} \uparrow$ | $A_{1.0} \uparrow$ | $A_{5.0} \uparrow$ | $A_{10.0} \uparrow$ |
| Base | 0.76 | 53.65 | 87.38 | 95.10 | 99.76 | 99.81 | 4.18 | 76.31 | 89.87 | **94.49** | 98.21 | 99.15 |
| PDCE | **0.63** | **55.63** | 88.04 | 95.24 | **99.86** | **99.91** | 3.89 | 76.64 | 89.97 | 94.35 | 98.21 | 99.20 |
| PDCE+MLN | 0.64 | 55.58 | 87.99 | 95.24 | 99.81 | **99.91** | 3.89 | **76.68** | **90.01** | 94.35 | 98.21 | 99.20 |
| PDCE+RAG | 0.65 | 55.49 | 88.79 | **95.34** | 99.81 | **99.91** | **3.85** | 76.31 | 89.68 | 94.07 | **98.30** | **99.25** |
| PDCE+MLN+RAG | 0.65 | 55.49 | **88.84** | **95.34** | 99.81 | **99.91** | **3.85** | 76.26 | 89.68 | 94.11 | **98.30** | **99.25** |

## C.2. Predicate Selection

Table 14 indicates that incorporating MLLM action predicates significantly enhances SafeAuto's effectiveness. Subsequently, we ranked all environmental predicates based on their total occurrence frequency across all scenarios. For each experiment, we selected the top $n$ ($n = 5, 10, 15, 20$) most frequently occurring environmental predicates, retrained the Markov Logic Network, and evaluated its performance. Notably, selecting only the top five environmental predicates achieved relatively high accuracy, suggesting that the majority of erroneous scenarios are associated with these predicates, such as SolidRedLight.

Table 14: Ablation study on the impact of different predicate selections on SafeAuto performance.

| Method | BDDX | | | | DriveLM | | | |
|---|---|---|---|---|---|---|---|---|
| | Acc $\uparrow$ | Speed $\uparrow$ | Steer $\uparrow$ | Avg. $\uparrow$ | Acc $\uparrow$ | Speed $\uparrow$ | Steer $\uparrow$ | Avg. $\uparrow$ |
| MLLM Action Predicates & All Environmental Predicates | **92.18** | **79.85** | 81.90 | **84.65** | **74.60** | **79.85** | 81.90 | 78.12 |
| All Environmental Predicates | 49.88 | 49.49 | 46.28 | 48.55 | 38.83 | 49.49 | 46.28 | 44.20 |
| MLLM Action Predicates & Top 5 Environmental Predicates | 87.75 | 79.27 | 81.75 | 82.26 | 74.16 | 79.27 | 81.75 | 78.39 |
| MLLM Action Predicates & Top 10 Environmental Predicates | 92.10 | 79.56 | 81.75 | 84.47 | 74.30 | 79.56 | 81.75 | 78.54 |
| MLLM Action Predicates & Top 15 Environmental Predicates | **92.18** | 79.42 | **81.90** | 84.50 | 74.31 | 79.42 | **81.90** | 78.54 |

## C.3. Rule Violation

Table 15 presents the impact of PDCE, RAG, and MLN on the violation of traffic rules in SafeAuto's action prediction. On the BDD-X dataset, PDCE and RAG significantly reduce the violation rate in the underlying MLLM's decision-making. The MLN post-verification further decreases the violation rate of SafeAuto. However, on the DriveLM dataset, PDCE and RAG do not reduce the violation rate of MLLM's prediction, as DriveLM contains a large number of simple driving scenes with only straight-line movement, making it challenging for MLLM to effectively learn different driving patterns. Nevertheless, using MLN to correct MLLM's errors reduces the violation rate.

Table 15: Ablation study on the impact of each module on the traffic rule violation rate of MLLM-predicted actions.

| Method | BDDX | DriveLM |
|---|---|---|
| Base | 11.64% | 1.03% |
| PDCE | 8.44% | 1.46% |
| RAG+PDCE | 5.90% | 1.03% |
| RAG+PDCE+MLN | **4.50%** | **0.75%** |

## D. Case Study

### D.1. High-Level Action Query

The two examples in Figure 5 use the base model and SafeAuto, respectively, to predict high-level actions. The actions predicted by SafeAuto are closer to the ground truth.

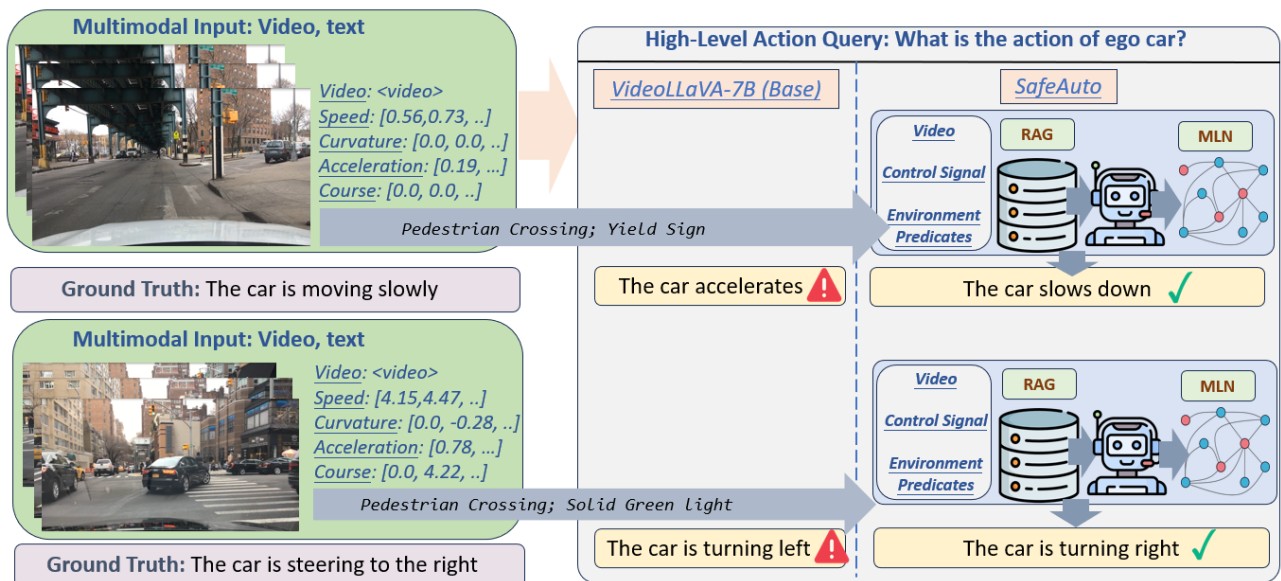

Figure 5: Case Study on High-Level Action Queries.

## D.2. Low-Level Action Query

Figure 6 shows that the low-level control signals obtained through SafeAuto for low-level action queries are numerically closer to the ground truth. Moreover, the predicted control signals are highly correlated with the previous high-level queries.

## D.3. Token Probability Distribution in Control Signal Prediction

Figure 7 illustrates the probability distribution of numerical tokens when predicting low-level control signals using the base model and SafeAuto.

## D.4. MLN Post-Verification

Figure 8 illustrates how the MLN rejects dangerous and illegal actions predicted by the MLLM and enhances safety by recommending high-level actions for the MLLM to re-output.

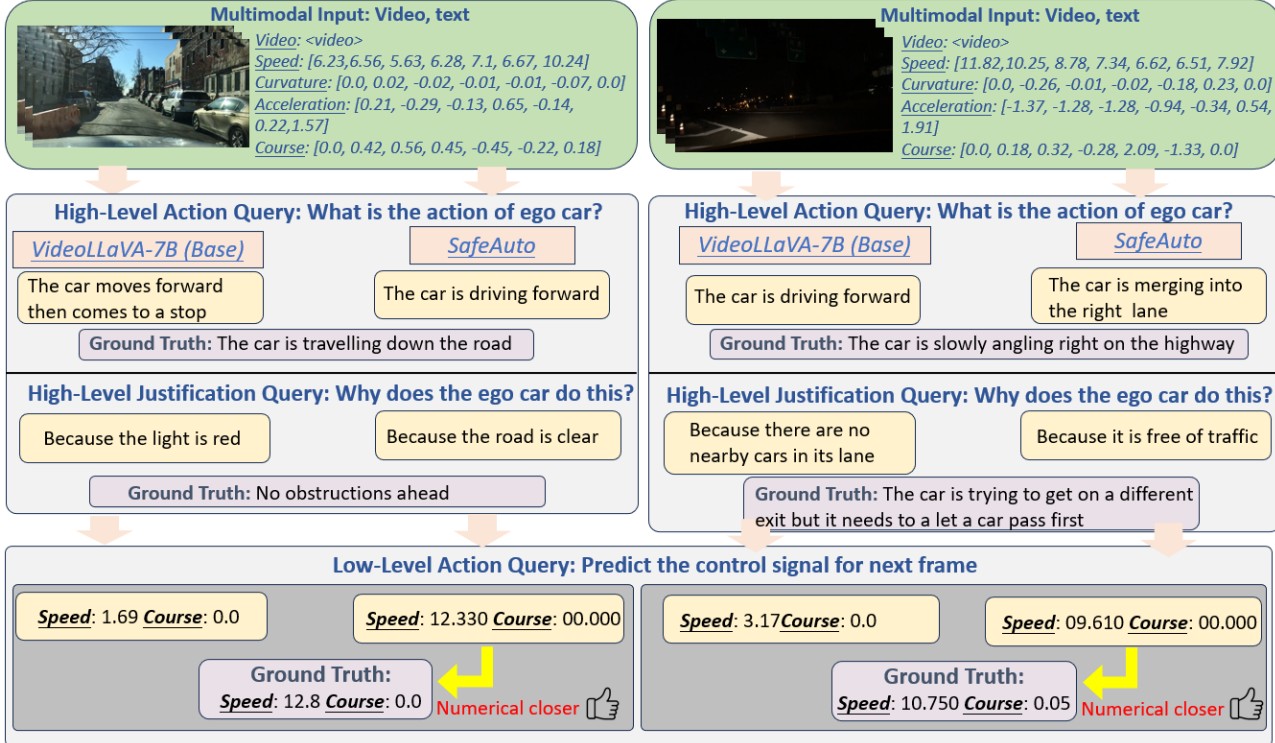

Figure 6: Case Study on Low-Level Action Queries.

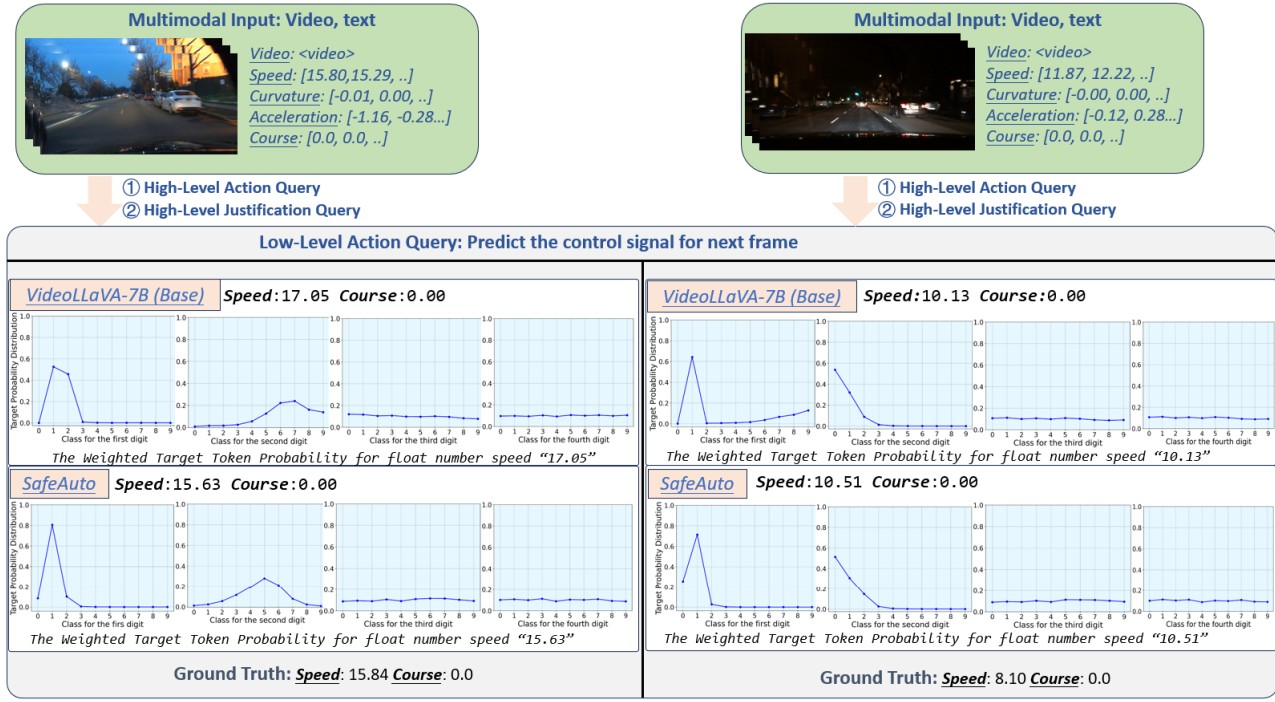

Figure 7: Case Study on Token Probability Distribution.

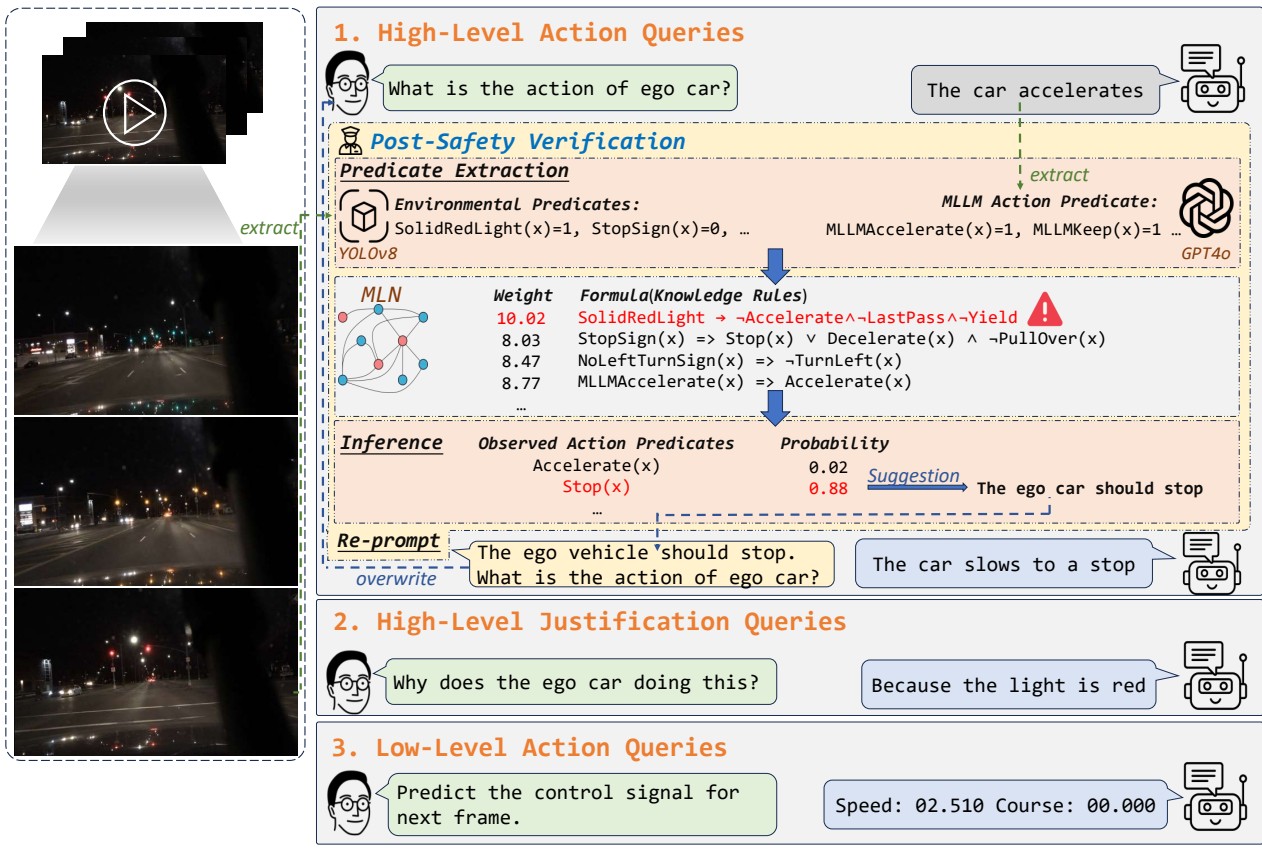

Figure 8: An example of rejecting and correcting aggressive behavior through MLN

---

**Predicates**

- *Unobserved Predicates:*
  `Keep(x)`, `Accelerate(x)`, `Decelerate(x)`, `Stop(x)`, `Reverse(x)`, `TurnLeft(x)`, `TurnRight(x)`, `UTurn(x)`, `Merge(x)`, `LeftPass(x)`, `RightPass(x)`, `Yield(x)`, `ChangeToLeftLane(x)`, `ChangeToRightLane(x)`, `Park(x)`, `PullOver(x)`
- *Observed Predicates:*
  - *MLLM Action Predicates:*
    `MLLMKeep(x)`, `MLLMAccelerate(x)`, `MLLMDecelerate(x)`, `MLLMStop(x)`, `MLLMReverse(x)`, `MLLMTurnLeft(x)`, `MLLMTurnRight(x)`, `MLLMUTurn(x)`, `MLLMMerge(x)`, `MLLMLeftPass(x)`, `MLLMRightPass(x)`, `MLLMYield(x)`, `MLLMChangeToLeftLane(x)`, `MLLMChangeToRightLane(x)`, `MLLMPark(x)`, `MLLMPullOver(x)`
  - *Environmental Predicates:*
    `SolidRedLight(x)`, `SolidYellowLight(x)`, `YellowLeftArrowLight(x)`, `RedLeftArrowLight(x)`, `MergingTrafficSign(x)`, `NoLeftTurnSign(x)`, `NoRightTurnSign(x)`, `PedCrossingSign(x)`, `StopSign(x)`, `RedYieldSign(x)`, `SlowSign(x)`, `SolidGreenLight(x)`, `HCSKeep(x)`, `HCSAccelerate(x)`, `HCSDecelerate(x)`, `HCSStop(x)`, `HCSReverse(x)`, `HCSStraight(x)`, `HCSLeft(x)`, `HCSRight(x)`

Table 16: The predicate set of BDD-X.

**Traffic Rules**

- *SolidRedLight(x) $\implies$ ¬Accelerate(x) ∧ ¬LeftPass(x) ∧ ¬Yield(x)*
- *SolidYellowLight(x) $\implies$ TurnLeft(x) ∨ TurnRight(x) ∨ Keep(x) ∨ Stop(x) ∨ Decelerate ∧ ¬Accelerate(x)*
- *YellowLeftArrowLight(x) $\implies$ Stop(x) ∨ Decelerate(x)*
- *RedLeftArrowLight(x) $\implies$ ¬(TurnLeft(x) ∨ UTurn(x))*
- *MergingTrafficSign(x) $\implies$ Decelerate(x)*
- *NoLeftTurnSign(x) $\implies$ ¬TurnLeft(x)*
- *NoRightTurnSign(x) $\implies$ ¬TurnRight(x)*
- *RedYieldSign(x) $\implies$ Decelerate(x)*
- *SlowSign(x) $\implies$ ¬Accelerate(x)*
- *StopSign(x) $\implies$ Stop(x) ∨ Decelerate(x) ∧ ¬PullOver(x)*
- *HCSKeep(x) $\implies$ Keep(x) ∨ Accelerate(x)*
- *HCSAccelerate(x) $\implies$ Keep(x) ∨ Accelerate(x)*
- *HCSDecelerate(x) $\implies$ Decelerate(x) ∨ Stop(x)*
- *HCSStop(x) $\implies$ Decelerate(x) ∨ Stop(x)*
- *HCSReverse(x) $\implies$ Reverse(x)*
- *HCSLeft(x) $\implies$ TurnLeft(x) ∨ ChangeToLeftLane(x)*
- *HCSRight(x) $\implies$ TurnRight(x) ∨ ChangeToRightLane(x)*
- *HCSLeft(x) ∧ MLLMChangeToRightLane(x) $\implies$ ChangeToLeftLane(x)*
- *HCSRight(x) ∧ MLLMChangeToLeftLane(x) $\implies$ ChangeToRightLane(x)*
- *MLLMKeep(x) $\implies$ Keep(x)*
- *MLLMAccelerate(x) $\implies$ Accelerate(x)*
- *MLLMDecelerate(x) $\implies$ Decelerate(x)*
- *MLLMStop(x) $\implies$ Stop(x)*
- *MLLMReverse(x) $\implies$ Reverse(x)*
- *MLLMTurnLeft(x) $\implies$ TurnLeft(x)*
- *MLLMTurnRight(x) $\implies$ TurnRight(x)*
- *MLLMUTurn(x) $\implies$ UTurn(x)*
- *MLLMMerge(x) $\implies$ Merge(x)*
- *MLLMLeftPass(x) $\implies$ LeftPass(x)*
- *MLLMRightPass(x) $\implies$ RightPass(x)*
- *MLLMYield(x) $\implies$ Yield(x)*
- *MLLMChangeToLeftLane(x) $\implies$ ChangeToLeftLane(x)*
- *MLLMChangeToRightLane(x) $\implies$ ChangeToRightLane(x)*
- *MLLMPark(x) $\implies$ Park(x)*
- *MLLMPullOver(x) $\implies$ PullOver(x)*

Table 17: First-order logic formulas of BDD-X.

**Possible Worlds**

*(Keep), (Accelerate), (Decelerate), (Stop), (TurnLeft), (TurnRight), (UTurn), (PullOver), (Reverse), (Park), (Merge), (LeftPass), (RightPass), (ChangeToLeftLane), (ChangeToRightLane), (Yield), (ChangeToRightLane, Merge), (Accelerate, ChangeToRightLane), (Decelerate, Stop), (Keep, Stop), (Accelerate, Keep), (Merge, Stop), (Accelerate, LeftPass), (ChangeToLeftLane, Merge), (Stop, Yield), (Accelerate, TurnRight), (Decelerate, Keep), (Decelerate, PullOver), (ChangeToLeftLane, PullOver), (ChangeToRightLane, Stop), (Keep, TurnRight), (PullOver, Stop), (Park, Stop), (Decelerate, TurnRight), (Keep, LeftPass), (Accelerate, ChangeToLeftLane), (Accelerate, TurnLeft), (Accelerate, Stop), (Keep, TurnLeft), (Accelerate, Merge), (Decelerate, TurnLeft), (Park, PullOver), (Keep, Merge), (Keep, Park), (TurnLeft, TurnRight), (TurnLeft, Reverse), (TurnRight, Stop), (ChangeToLeftLane, Decelerate), (ChangeToRightLane, Decelerate), (TurnLeft, Stop), (TurnRight, Park), (ChangeToLeftLane, ChangeToRightLane), (Keep, RightPass), (ChangeToLeftLane, Stop), (Keep, PullOver), (LeftPass, RightPass), (ChangeToRightLane, Keep), (TurnRight, PullOver), (ChangeToLeftLane, Keep), (TurnRight, Reverse), (PullOver, Reverse), (ChangeToRightLane, TurnLeft), (Accelerate, Decelerate), (TurnRight, Yield), (Decelerate, Yield), (ChangeToRightLane, PullOver), (TurnLeft, PullOver), (Decelerate, TurnLeft, Stop), (Decelerate, Merge, Stop), (Decelerate, PullOver, Stop), (ChangeToRightLane, Decelerate, Stop), (ChangeToLeftLane, Decelerate, Stop), (Decelerate, TurnRight, Stop), (Accelerate, ChangeToLeftLane, ChangeToRightLane), (ChangeToRightLane, Decelerate, Merge), (ChangeToRightLane, Decelerate, Merge, Stop)*

Table 18: The possible worlds of BDD-X.

**Predicates**

- *Unobserved Predicates:*
  `Normal(x),Fast(x),Slow(x),Stop(x),Left(x),Right(x),Straight(x)`
- *Observed Predicates:*
    - *MLLM Action Predicates:*
      `MLLMNormal(x)`, `MLLMFast(x)`, `MLLMSlow(x)`, `MLLMStop(x)`, `MLLMLeft(x)`, `MLLMRight(x)`, `MLLMStraight(x)`
    - *Environmental Predicates:*
      `SolidRedLight(x)`, `SolidYellowLight(x)`, `YellowLeftArrowLight(x)`, `RedLeftArrowLight(x)`, `MergingTraffic(x)`, `NoLeftTurnSign(x)`, `NoRightTurnSign(x)`, `PedCrossingSign(x)`, `StopSign(x)`, `RedYieldSign(x)`, `SlowSign(x)`, `SolidGreenLight(x)`, `DoubleDashedWhiteLineLeft(x)`, `DoubleDashedWhiteLineRight(x)`,`SingleSolidWhiteLineLeft(x)`, `SingleSolidWhiteLineRight(x)`,`DoubleSolidWhiteLineLeft(x)`, `DoubleSolidWhiteLineRight(x)`,`SingleZigzagWhiteLineLeft(x)`, `SingleZigzagWhiteLineRight(x)`,`SingleSolidYellowLineLeft(x)`, `SingleSolidYellowLineRight(x)`, `HCSNormal(x)`, `HCSFast(x)`, `HCSSlow(x)`, `HCSStop(x)`, `HCSLeft(x)`,`HCSRight(x)`,`HCSStraight(x)`

Table 19: The predicate set of DriveLM.

**Traffic Rules**

- `SolidRedLight(x) ⟹ ¬Fast(x)`
- `SolidYellowLight(x) ⟹ ¬Fast(x)`
- `YellowLeftArrowLight(x) ⟹ Stop(x)∨Slow(x)`
- `RedLeftArrowLight(x) ⟹ ¬Left(x)`
- `MergingTrafficSign(x) ⟹ ¬Fast(x)`
- `NoLeftTurnSign(x) ⟹ ¬Left(x)`
- `NoRightTurnSign(x) ⟹ ¬Right(x)`
- `RedYieldSign(x) ⟹ ¬Fast(x)`
- `SlowSign(x) ⟹ ¬Fast(x)`
- `SingleSolidWhiteLineLeft(x) ⟹ ¬Left(x)`
- `SingleSolidWhiteLineRight(x) ⟹ ¬Right(x)`
- `DoubleSolidWhiteLineLeft(x) ⟹ ¬Left(x)`
- `DoubleSolidWhiteLineRight(x) ⟹ ¬Right(x)`
- `SingleZigzagWhiteLineLeft(x) ⟹ ¬Stop(x)`
- `SingleZigzagWhiteLineRight(x) ⟹ ¬Stop(x)`
- `HCSNormal(x) ⟹ Normal(x)`
- `HCSFast(x) ⟹ Fast(x)`
- `HCSSlow(x) ⟹ Slow(x)`
- `HCSStop(x) ⟹ Stop(x)`
- `HCSLeft(x) ⟹ Left(x)`
- `HCSRight(x) ⟹ Right(x)`
- `HCSStraight(x) ⟹ Straight(x)`
- `MLLMNormal(x) ⟹ Normal(x)`
- `MLLMFast(x) ⟹ Fast(x)`
- `MLLMSlow(x) ⟹ Slow(x)`
- `MLLMStop(x) ⟹ Stop(x)`
- `MLLMLeft(x) ⟹ Left(x)`
- `MLLMRight(x) ⟹ Right(x)`
- `MLLMStraight(x) ⟹ Straight(x)`

Table 20: First-order logic formulas of DriveLM.

**Possible Worlds**

*(Normal, Left), (Normal, Right), (Normal, Straight), (Fast, Left), (Fast, Right), (Fast, Straight), (Slow, Left), (Slow, Right), (Slow, Straight), (Stop, Left), (Stop, Right), (Stop, Straight),*

Table 21: The possible worlds of DriveLM.

