# OpenReview forum: "SafeAuto: Knowledge-Enhanced Safe Autonomous Driving with Multimodal Foundation Models"
_ICML.cc/2025/Conference — ICML 2025 poster_

### Official Review · Reviewer_KjQv · 2025-03-02

**Overall Recommendation:** 4

**Summary:**

This paper presents SafeAuto, an MLLM-based autonomous driving system. SafeAuto has three major innovations. First, it uses a new Position-Dependent CE loss (PDCE) loss, which supervises the predicted number tokens based on their numerical difference from the ground-truth number. Second, it has a knowledge-enhanced post-safety verification module using Markov Logic Networks (MLNs). It can explicitly encode domain knowledge and structured traffic rules into the decision-making process of the MLLM, which can be used to verify and correct the predicted high-level actions. Third, it uses a novel training method for constructing a unified embedding that effectively integrates all modalities.

The authors evaluated the SafeAuto method on the BDD-X dataset and the DriveLM dataset. They showed that SafeAuto has better driving performance than the state-of-the-art baselines.


## update after rebuttal

No updates.

**Claims And Evidence:**

* The proposed PDCE loss and MLN-based reasoning module are well-designed
* The overall SafeAuto method shows strong performance against state-of-the-art baselines on popular benchmarks.

**Essential References Not Discussed:**

* I don't have anything to add.

**Experimental Designs Or Analyses:**

* The experimental setup is valid. However, I think all the results are from open-loop simulation. I would like to see some results in closed-loop simulation as well.

**Methods And Evaluation Criteria:**

* The datasets used for evaluation make sense.

**Other Comments Or Suggestions:**

* The motivation for the PDCE loss is that using the MSE loss on regression outputs will "disrupt the MLLM’s autoregressive token generation, transforming it into a pure transformer encoder (Tan et al., 2024) used only for regression tasks and losing its language generation capabilities necessary for high-level question-answering". I hope the authors can compare their method against this MSE-loss baseline and show how the high-evel question-answer capabilities are impacted.

* Figure 2. Having two peaks is not necessarily worse than having just one peak. It could be that there are two different valid maneuvers in this scenario. It will be useful to have more analysis on why the traditional CE loss leads to the observed behavior.

* It is not clear how SafeAuto leverages past experiences to inform current decision-making and how Multimodal RAG helps.

**Other Strengths And Weaknesses:**

N/A

**Questions For Authors:**

See "Other Comments or Suggestions".

**Relation To Broader Scientific Literature:**

N/A

**Theoretical Claims:**

N/A

---

> ### Author Rebuttal · Authors · 2025-04-01
>
> We extend our sincere gratitude to the reviewer for their meticulous and constructive feedback. Their insightful observations and valuable recommendations have greatly contributed to improving the rigor and clarity of our work!
>
> > **Q1: The motivation for the PDCE loss is that using the MSE loss on regression outputs will "disrupt the MLLM’s autoregressive token generation, transforming it into a pure transformer encoder (Tan et al., 2024) used only for regression tasks and losing its language generation capabilities necessary for high-level question-answering". I hope the authors can compare their method against this MSE-loss baseline and show how the high-level question-answer capabilities are impacted.**
>
> Thank you for the valuable suggestion! In fact, once we apply an MSE loss at the MLLM’s hidden layer to directly regress the low-level control signals, the model stops generating tokens—only numeric outputs are produced via an MLP head, rather than text tokens, and thus we cannot do a further next-token generation. Consequently, it loses its autoregressive language generation capability, making further high-level QA impossible. As a result, the TimeLLM can only handle low-level predictions but cannot generate further textual answers or explanations.
>
> Moreover, even when focusing exclusively on low-level tasks, our proposed SafeAuto still outperforms TimeLLM by a wide margin on BDD-X dataset, as illustrated in the tables below:
>
> Speed:
>
> | Method       | RMSE     | A0.1      | A0.5      | A1.0      | A5.0      | A10.0     |
> | ------------ | -------- | --------- | --------- | --------- | --------- | --------- |
> | TimeLLM      | 1.17     | 21.34     | 53.13     | 74.14     | 99.67     | 99.86     |
> | **SafeAuto** | **0.65** | **55.49** | **88.84** | **95.34** | **99.81** | **99.91** |
>
> Course:
>
> | Method       | RMSE     | A0.1      | A0.5      | A1.0      | A5.0      | A10.0     |
> | ------------ | -------- | --------- | --------- | --------- | --------- | --------- |
> | TimeLLM      | 4.10     | 65.70     | 83.47     | 89.59     | 97.60     | 98.59     |
> | **SafeAuto** | **3.85** | **76.26** | **89.68** | **94.11** | **98.30** | **99.25** |
>
>
>
> > **Q2: Figure 2. Having two peaks is not necessarily worse than having just one peak. It could be that there are two different valid maneuvers in this scenario. It will be useful to have more analysis on why the traditional CE loss leads to the observed behavior.**
>
> Thank you for the insightful question! In that case, the historical speed values are [6.23, 6.56, 5.62, 6.27, 7.09, 6.67, 10.24] with the ground truth being 12.46. This means that the peak around 13 is actually more valid in this scenario. But we agree that in some cases, two peaks might indicate different valid maneuvers. However, since we are predicting for the very next frame—a short time interval—the two peaks should not be widely separated as shown in Fig 2. In such cases, we would expect the distribution to resemble a Gaussian with one dominant peak more.
>
> The issue with CE loss lies in how it computes the joint probability. The calculation follows p('1') × p('2'|'1') × p('4'|'12') × p('6'|'12.4'), treating every digit as equally important during training. Thus, when the training data is not large enough, the model might learn to assign higher values to the last three digits instead of the first digit to make the overall probability high. As a result, even if the latter probabilities are high, a lower value for p('1') can cause the model to generate a different starting digit (for instance, '3' instead of '1'), which then leads to the subsequent digits being generated incorrectly.
>
> We also provide two case studies in Figure 6 on Page 16 to illustrate these failure cases with CE loss more clearly. We will also include more additional analysis in our final version to further explain this behavior!
>
> > **Q3: It is not clear how SafeAuto leverages past experiences to inform current decision-making and how Multimodal RAG helps.**
>
> Sorry for the confusion, to clarify, once we retrieve similar past experiences, we incorporate the corresponding historical video along with both high-level and low-level predictions into the current context—that is, we append them before the current video and question in prompt. This process is similar to few-shot learning because it provides the MLLM with additional context from similar driving scenarios, thereby informing its current predictions. We will provide a more detailed explanation in our final version.
>
> If you have any further questions or suggestions, please feel free to let us know. Your feedback is greatly appreciated and will certainly help us improve our work a lot!

---

> > ### Comment · Reviewer_KjQv · 2025-04-04
> >
> > Thank you for your clarifications.

---

### Official Review · Reviewer_v68f · 2025-03-10

**Overall Recommendation:** 3

**Summary:**

This paper proposes SafeAuto, a novel framework for autonomous driving using multimodal foundation models. It addresses the challenges of integrating high-level reasoning and low-level control.
The main algorithm ideas include three key components. First, the Position-Dependent Cross-Entropy (PDCE) loss function improves low-level numerical prediction accuracy while maintaining the autoregressive nature of the Multimodal Large Language Model (MLLM). Second, Knowledge-Enhanced Post-Safety Verification uses Markov Logic Networks (MLNs) to integrate traffic rules into the MLLM's decision-making process, verifying and correcting high-level actions. Third, Multimodal Retrieval Augmented Generation (RAG) learns from similar driving experiences by integrating video data, control signals, and environmental predicates.
The paper's main findings show that SafeAuto outperforms existing baselines. On the BDD-X and DriveLM datasets, it reduces the Root Mean Square Error (RMSE) for speed and course predictions and improves high-level action prediction performance. On the BDD-X dataset, it reduces the RMSE for speed and course predictions by 5.8% and 14.1% respectively, and boosts high-level action performance by 28.0% under the CIDEr metric.

## Update after rebuttal
Thanks for authors to provide answers that address all my concerns. I've also read the comments from the other reviewers. In overall consideration, I'd like to raise my assessment score to be positive.

**Claims And Evidence:**

The paper claims that SafeAuto, with its novel components like PDCE loss, MLN-based safety verification, and Multimodal RAG, outperforms existing baselines. This is supported by experiments on BDD-X and DriveLM datasets. For example, it shows significant improvements in low-level control accuracy, such as reducing RMSE for speed and course predictions, and high-level action prediction performance, like boosting CIDEr scores. Ablation studies are provided to help in understanding the contribution of each component.

**Essential References Not Discussed:**

There seem no missing essential related works.

**Experimental Designs Or Analyses:**

For the PDCE loss experiments, using the BDD-X dataset to compare its performance against the original CE loss is valid.
Measuring RMSE for speed and course predictions directly assesses the improvement in numerical prediction accuracy.
The MLN-based safety verification experiments are well-designed in terms of using real and simulated data for training.
The Multimodal RAG experiments use appropriate datasets and metrics.

**Methods And Evaluation Criteria:**

The proposed methods and evaluation criteria are suited for autonomous driving. The PDCE loss, MLN-based safety verification, and Multimodal RAG directly address the challenges of integrating high-level reasoning and low-level control.
Benchmark datasets like BDD-X and DriveLM, along with relevant metrics such as RMSE and CIDEr, comprehensively assess system performance in both high-level action prediction and low-level control accuracy, making them appropriate for this application.

**Other Comments Or Suggestions:**

See questions for authors.

**Other Strengths And Weaknesses:**

Strengths: The paper introduces concepts like PDCE loss, use of MLNs and Multimodal RAG. Experimental results shows that by integrating the designs, SafeAuto outperforms existing baselines across multiple datasets.

Weaknesses:
- Despite of the integration to form the SafeAuto pipeline, the novelty for each component seems marginal, which is my major concern.
- The writing is lengthy, and it seems some parts are overlength such as the abstract section.
- The arrangement of space between texts and figures need to be checked.

**Questions For Authors:**

Despite of the integration to form the SafeAuto pipeline, the novelty for each component seems not clear.
(1) The PDCE is weighted sum of the KLdivergence, what is new in this loss design?
(2) What is the difference between this work and the Markov Logic Networks (MLNs) of Richardson& Domingos, 2006?
(3) What are the special challenges addressed in the multimodal RAG method design?

**Relation To Broader Scientific Literature:**

Existing methods using MLLMs struggle with low-level control and safety. SafeAuto's PDCE loss improves low-level prediction accuracy compared to traditional cross-entropy loss. Its use of MLNs for safety verification addresses the lack of explicit safety checks in previous approaches. Multimodal RAG enhances decision-making by leveraging past experiences.

**Theoretical Claims:**

The paper focuses more on empirical validation through experiments instead of theoretical proofs.

---

> ### Author Rebuttal · Authors · 2025-04-01
>
> We sincerely thank the reviewer for their thoughtful and constructive feedback, and for recognizing the value of our work! The insightful suggestions and detailed comments provided have substantially contributed to enhancing the quality of our work!
>
> > **Q1: The PDCE is the weighted sum of the KL divergence, what is new in this loss?**
>
> Thanks for the insightful question! Our motivation for proposing the PDCE loss for LLM digit prediction is twofold. (1) When fine-tuning an LLM for digit prediction, one natural idea is to use a combination of CE loss for the word part and MSE loss for the digit part (which we have also tried internally before). However, this approach introduces some challenges: it is challenging to balance the magnitudes of the CE and MSE losses, the self-attention mechanism injects noise into the final output features for forecasting digits, and applying MSE loss on hidden features would also impair the LLM's autoregressive capability (i.e., it cannot do the next-token generation). (2) Additionally, using CE loss as existing work directly on digit prediction treats each digit as equally important, which introduces bias as highlighted in Figure 2.
>
> To address these challenges, we introduce the PDCE loss, which employs a careful weighted sum of the KL divergence. However, note that as shown in Fig 3, both the weights and the target distribution in the KL divergence are not randomly chosen, but determined formally by a predefined soft target distribution, such as a Gaussian distribution, which aims to approximate the behavior of the MSE loss on digits in string form. In this way, it solves the above challenges and eliminates the need to balance disparate losses for finetuning LLM, preserves the autoregressive property of the LLM, and also emulates the behavior of MSE loss during training on float numbers in string, as demonstrated in Figure 2.
>
> We also present a performance comparison with TimeLLM, which directly applies MSE loss on the extracted hidden features from LLM for regression, as suggested by Reviewer KjQv, below, and our method still performs much better. Due to the word limit, we kindly encourage you to review our rebuttal in Q1 for Reviewer KjQv.
>
> > **Q2: What is the difference between this work and the Markov Logic Networks (MLNs) of Richardson & Domingos, 2006?**
>
> Thank you for your insightful question! The key distinction in our work lies in how we integrate them into a data-driven vision-language model framework for autonomous driving.
>
> Specifically:
>
> 1. **Integration with Data-Driven Paradigms:** Our approach is the first to combine MLNs with Multimodal LLM in the context of autonomous driving. This integration allows us to inject explicit safety knowledge—such as traffic rules—into the prediction process. Directly training a model on data alone does not guarantee adherence to these safety constraints.
>
> 2. **Different Source of Predicates:** In our work, the grounding of predicates comes from multiple external modules. We incorporate outputs from our self-trained YOLOv8 detector and the original predictions from the MLLM to form the predicates.
>
> 3. **Joint Simulated and Real-Data Training:** Another novel aspect of our work is the joint training strategy for the MLN weights using both simulated and real-world data instead of purely relying on real-world data which may be hard to collect in practice.
>
> > **Q3: What are the special challenges addressed in the M-RAG?**
>
> Thank you for the insightful question! In contrast to relying solely on the video modality (as in RAGDriver), our Multimodal RAG must integrate three key modalities—(i) video or image, (ii) control signals, and (iii) environmental predicates—into a single unified embedding for ranking and retrieval. One challenge here is the lack of a clear objective for aligning these diverse modalities in autonomous driving retrieval tasks.
>
> To address this gap, we exploit textual scenario descriptions, since they naturally encapsulate information about all modalities. During training, our approach guides the unified embedding to replicate the relative ranking derived from these textual embeddings. Concretely, we use a contrastive learning-based procedure where each batch’s embedding distribution is pushed to match the ranking distribution from the text embedding for that scenario. This approach is both efficient (as it avoids iterating over external databases mid-training) and effective at aligning multiple modalities.
>
> By contrast, methods such as RAGDriver use a triplet-loss formulation, which only focuses on top-k similarities but ignores fine-grained ranking signals and requires pre-fetching positive/negative examples before training. The improved performance results also highlight the effectiveness of our proposed multimodal RAG, and as demonstrated in Table 10 on page 15, we find incorporating environmental predicates significantly boosts retrieval accuracy by mitigating some of the noise present in raw video embeddings.

---

### Official Review · Reviewer_EhoY · 2025-03-14

**Overall Recommendation:** 3

**Summary:**

This paper proposes SafeAuto, a novel framework that enhances MLLM-based autonomous driving systems by incorporating both unstructured and structured knowledge. The model can predict high-level and low-level action prediction.

**Claims And Evidence:**

Yes.

**Essential References Not Discussed:**

No other essential references are not discussed.

**Experimental Designs Or Analyses:**

The experiments are convincing and show the effectiveness of each component. However, the experiments are only conducted on the BDD-X/DriveLM datasets and can only be compared with some narrow works.

**Methods And Evaluation Criteria:**

Yes. However, the proposed method uses Weighted Target Token Probability to predict the float number of speed/course, somehow not a very accurate predicting approach for these fine-grained signals.

**Other Comments Or Suggestions:**

The writing is not very friendly for those readers in the autonomous driving field. I recommend adding relevant background information about the methods.

**Other Strengths And Weaknesses:**

Strengths:
- This paper describes the details of the method very clearly. Although it takes some effort to follow, the proposed method is very systematic.

Weaknesses:
- The experiments are only conducted on the BDD-X/DriveLM datasets and may only be compared with some narrow works.
- The proposed method uses Weighted Target Token Probability to predict the float number of speed/course, somehow not a very accurate predicting approach for these fine-grained signals.
- I have some doubts about whether the proposed method is very effective for low-level actions. The BDD-X dataset is not a universally recognized best dataset for evaluating low-level actions. Maybe it needs to be compared with some end-to-end autonomous driving works.

**Questions For Authors:**

Please refer to the weakness mentioned above.

**Relation To Broader Scientific Literature:**

This paper provides a Knowledge-Enhanced large model for autonomous driving to predict high-level and low-level actions.

**Theoretical Claims:**

NA.

---

> ### Author Rebuttal · Authors · 2025-04-01
>
> We are deeply grateful to the reviewer for their thorough and insightful feedback. Their expertise and dedicated time have significantly contributed to improving the quality of this work!
>
>
> > **Q1: The experiments are only conducted on the BDD-X/DriveLM datasets and may only be compared with some narrow works.**
>
> Thank you for the valuable suggestion. We fully recognize the need to test our framework on a broader range of datasets to better demonstrate its generalizability and effectiveness. Unfortunately, the availability of comprehensive multimodal autonomous driving datasets is currently limited. For example, nuScenes, while extensive, lacks high-level action or reasoning annotations (and DriveLM is just developed based on nuScenes which provides the high-level annotation). Similarly, the Waymo dataset does not provide essential modality data like images or videos. Given these constraints, BDD-X and DriveLM represent the most relevant and widely used datasets for evaluating multimodal autonomous vehicle systems involving LLMs. We hope that the future development of diverse multimodal datasets will allow for more comprehensive evaluations of frameworks like ours.
>
> > **Q2: The proposed method uses Weighted Target Token Probability to predict the float number of speed/course, somehow not a very accurate predicting approach for these fine-grained signals.**
>
> Thank you for your insightful question! As detailed in our paper, our PDCE loss yields notably higher accuracy on these fine-grained signals. For example, on the DriveLM dataset—where the goal is to predict the next six waypoints—our method achieves an ADE of 0.84, substantially lower than the 1.51 reported for DriveLM-agent and even approaching the performance of the full regression model UniAD (full) with ADE as 0.80. Additionally, on the BDD-X dataset, and following reviewer KjQv’s suggestion, our approach still outperforms TimeLLM, which employs an MSE loss on LLM-extracted hidden features. The performance comparison for speed prediction is summarized below:
>
> Speed:
>
> | Method       | RMSE     | A0.1      | A0.5      | A1.0      | A5.0      | A10.0     |
> | ------------ | -------- | --------- | --------- | --------- | --------- | --------- |
> | TimeLLM      | 1.17     | 21.34     | 53.13     | 74.14     | 99.67     | 99.86     |
> | **SafeAuto** | **0.65** | **55.49** | **88.84** | **95.34** | **99.81** | **99.91** |
>
> Course:
>
> | Method       | RMSE     | A0.1      | A0.5      | A1.0      | A5.0      | A10.0     |
> | ------------ | -------- | --------- | --------- | --------- | --------- | --------- |
> | TimeLLM      | 4.10     | 65.70     | 83.47     | 89.59     | 97.60     | 98.59     |
> | **SafeAuto** | **3.85** | **76.26** | **89.68** | **94.11** | **98.30** | **99.25** |
>
> So it is indeed accurate and the weights here are also not arbitrarily chosen but determined by a predefined soft target distribution. Moreover, the ablation study on the hyperparameter $\sigma$, as shown in Fig. 4 on page 15, further validates the robustness and effectiveness of this weighting strategy.
>
> > **Q3: I have some doubts about whether the proposed method is very effective for low-level actions. The BDD-X dataset is not a universally recognized best dataset for evaluating low-level actions. Maybe it needs to be compared with some end-to-end autonomous driving works.**
>
> Thank you for your insightful question! In our experiments, we also used the DriveLM dataset, which is derived from the standard nuScenes dataset and specifically designed for end-to-end prediction. The goal is to forecast 3 seconds into the future (i.e., 6 future waypoints), thereby providing a standard framework for low-level action evaluation. As demonstrated in Table 2, our method still achieves substantially improved low-level action prediction, with an ADE of 0.84 compared to the 1.51 ADE reported for the original DriveLM. This performance is also on par with the full regression method UniAD (full), which records an ADE of 0.80. Currently, there are few multimodal autonomous driving datasets that include high-level annotations. Nevertheless, we are still excited to evaluate our method on any new datasets offering comprehensive multimodal annotations for both high-level and low-level actions as they become available!
>
> > **Q4: The writing is not very friendly for those readers in the autonomous driving field. I recommend adding relevant background information about the methods.**
>
> Thank you for your valuable suggestion! We acknowledge that the current version may not be fully accessible to readers in the autonomous driving field. We will add more background details, such as for RAG, in the related works section to better support these readers.

---

### Official Review · Reviewer_X7JM · 2025-03-16

**Overall Recommendation:** 3

**Summary:**

SafeAuto proposes a unified framework to enhance autonomous driving systems by leveraging multimodal foundation models. It integrates three core components:

Position-Dependent Cross-Entropy (PDCE) Loss: An adaptation of the standard cross-entropy loss that incorporates digit-level proximity and place-level weighting, making it behave more like MSE for low-level numerical predictions.

Knowledge-Enhanced Post-Safety Verification: Utilizes a Markov Logic Network (MLN) to explicitly encode traffic rules and safety constraints, serving as a verification layer that can override unsafe high-level action predictions from the multimodal language model.

Multimodal Retrieval-Augmented Generation (RAG): Combines video data, control signals, and environmental predicates into a unified embedding space to retrieve similar driving experiences and inform both high-level and low-level decision making.

The proposed methods are evaluated on datasets BDD-X and DriveLM, which shows improvements in both low-level control signal accuracy (e.g., reductions in RMSE for speed and course) and high-level behavior prediction (except justification).

**Claims And Evidence:**

The paper claims that by integrating a modified loss function, explicit safety verification via MLN, and a multimodal retrieval mechanism, SafeAuto significantly improves both low-level control precision and high-level decision-making in autonomous driving.

The work is generally well-motivated and the PDCE loss is especially targeted to improve the numeric prediction ability for autonomous driving tasks. The evaluations also reflect the effectiveness of this design - the prediction accuracy increases (especially the low-level motion/control signal prediction). The MLN is intended to improve the safety and the RAG is meant to enhance the reasoning. The ablation study also shows how different modules contribute to the final improvement. However, in the experiments, it would be great if the authors can provide more detailed results on safety-critical scenarios (besides the rule-following). Besides, the justifications scores on the BDD-X data is behind all baselines and it is worth explaining and further investigation. How the RAG datasets are built and how feasible the RAG is are still unclear, especially given the significant improvement brought by the RAG module.

**Essential References Not Discussed:**

The paper includes the recent advances in MLLMs for autonomous driving and the safety guarantees for such systems. However, I also encourage the authors to include the works on enhancing safety for general LLM-based autonomous systems because the system-level design of MLLM-based and general LLM-based AV systems are similar. For instance, the work 'DiLu: A Knowledge-Driven Approach to Autonomous Driving with Large Language Models' (ICLR 24) utilize a similar memory module to enhance the performance and reasoning ability. The works 'Empowering Autonomous Driving with Large Language Models: A Safety Perspective' (ICLRW 2024) discussed a couple methods with verification modules and in-context learning to enhance the safety.

**Experimental Designs Or Analyses:**

As mentioned above, the paper provide extensive results/ablation study on both high-level and low level motion prediction/decision-making, which demonstrates the effectiveness of the proposed methods.

However, there are still some concerns regarding the evaluation:

1) the justification score falls behind all baselines and I didn't find convincing explanation or ablation for that - the explainability is one of the key motivation why we use the (M)LLM for such safety-critical tasks. If we only cares about the motion prediction/action decision-making, LLM-based methods may not be necessary or efficient.

2) If we look into the detailed ablation study, we find for most metrics the RAG brings most improvement, especially for the high-level predictions. It is unclear how the RAG dataset is built and how similar they are to the actual test data? This can be critical to understand the feasibility and generalizability of the proposed methods.

**Methods And Evaluation Criteria:**

The authors provide extensive results on two major multi-modal driving dataset BDD-x and DriveLM and they also presents the results on both high-level and low-level prediction/decision-making, which support most of their claimed contribution.

Generally one concern of the (m)LLM-based is how efficient the inference process is - although the paper mentions the inference is computationally efficiently. It would be great if the author can provide more details on that.

**Other Comments Or Suggestions:**

Please kindly refer to the previous comments.

**Other Strengths And Weaknesses:**

* Strength:

The PDCE loss is a creative solution to the problem of numerical prediction with autoregressive models, retaining language generation abilities while improving regression performance.

The explicit incorporation of traffic rules using MLNs directly addresses the safety-critical requirements of autonomous driving.

The framework is evaluated on two datasets with both qualitative and quantitative results, and ablation studies clarify the contributions of each component.

* Weakness:

Hyperparameter Sensitivity: The performance of PDCE loss and the multimodal RAG component depends on carefully chosen parameters (e.g., σ, weighting factors). A deeper exploration of this sensitivity would be beneficial.

Reliance on External Modules: The approach relies on pretrained object detectors (YOLOv8) and text encoders, making it vulnerable to errors from these components.

Limited Domain Evaluation: Although evaluated on two datasets, additional tests in diverse driving scenarios or simulated environments could further demonstrate the generalizability and robustness of SafeAuto. LLM's strength for AD tasks is its general common sense and generalizability.

**Questions For Authors:**

Please kindly refer to previous comments/questions

**Relation To Broader Scientific Literature:**

The paper is based on the recent advance of (M)LLMs and a series of pioneering works on utilizing LLMs to autonomous driving tasks. They focus on improving loss design and enhancing the safety of such syetems.

**Theoretical Claims:**

The theoretical contribution mainly lie on the design of PDCE loss, which makes sense to me. I believe it helps mitigate the LLM' intrinsic issue of poor numeric reliability.

The theoretical framework for using MLNs to encode traffic rules is solid, but its practical performance depends on accurate predicate extraction and the proper setting of rule weights. The paper could delve deeper into how uncertainties in rule extraction might affect inference in the MLN.

---

> ### Author Rebuttal · Authors · 2025-04-01
>
> We are deeply grateful to the reviewer for their insightful and thorough feedback, and we appreciate the recognition of our work's contribution! The suggestions and comments made for our work have significantly helped to improve its quality.
>
> > **Q1: Could the authors provide more details on the inference efficiency of the proposed (M)LLM-based method?**
>
> Thanks for raising this insightful question! We employ KV-caching to accelerate the MLLM’s inference during multi-turn conversations, enabling faster and more efficient responses. Using a single NVIDIA A6000 GPU in a standard academic setting, we recorded an average inference time of approximately 2.33 seconds per case with the full pipeline (i.e., including RAG, covering both high-level and low-level predictions with Safe-Reasoning) on the BDD-X dataset, and 3.50 seconds per case on the DriveLM dataset. In our practical industry deployments, the inference speed can be improved by 2–3 times. We will incorporate more detailed results on the inference efficiency into our final version.
>
> > **Q2: Why are the justification scores lower compared to baselines?**
>
> Thanks for your insightful observation. While our justification score appears lower relative to some baselines, it's substantially improved when compared to the vanilla base model without any components (see Table 11 on Page 15). While for other baselines, ADAPT is not an LLM-based method and independently predicts justifications separate from low-level actions. Compared to DriveGPT, our justification scores are comparable, but we significantly outperform it in high-level action prediction accuracy. Regarding RAGDriver, it uses both train and test data for training RAG, which we believe introduces evaluation bias; while our experiments adhere strictly to using only the training data. Notably, when adopting the setting of RAGDriver, actually we will have 50% further improvement in justification scores, but we don’t think it is a correct setting for evaluation.
>
> > **Q3: Could the authors provide more details on how the RAG dataset was constructed and clarify its similarity to the test data, given its significant impact on performance?**
>
> Thanks for your thoughtful question! Our RAG dataset is built exclusively from training data without incorporating any information from the test set. Upon manual inspection, we observed they are not extremely similar, suggesting the model generalizes effectively rather than relying on direct memorization. The superior performance of SafeAuto over RAGDriver also highlights the efficiency and generalizability of our RAG design.
>
> > **Q4: Essential References Not Discussed**
>
> Thanks for the helpful references! We have now cited these works and expanded the related discussion in our new version accordingly.
>
> > **Q5: Could the authors provide insights into the sensitivity of performance to hyperparameters in the PDCE loss and multimodal RAG component?**
>
> Thank you for highlighting this important point, and we apologize for the confusion. We indeed conducted sensitivity analyses for key hyperparameters, but the corresponding results are deferred to Appendix C. Specifically, Figure 4 on Page 15 shows our ablation study on the weighting factor $\sigma$. The results indicate that our PDCE loss could still consistently outperform the CE loss across a wide range of different $\sigma$ values, demonstrating robustness to hyperparameter selection.
>
> > **Q6: Could the authors discuss potential vulnerabilities arising from reliance on external modules such as pretrained detectors (YOLOv8) and text encoders?**
>
> Thanks for your insightful observation! While we acknowledge that external modules like YOLOv8 and pretrained text encoders can introduce errors—a common limitation shared by related works (e.g., AgentDriver). However, as we incorporate post-hoc safety verification through our SafeAuto-Reasoning component, it can correct certain errors originating from these modules and thus enhance driving stability. The substantial overall improvements obtained by leveraging these external modules also indicate that they provide more benefits than drawbacks.
>
> > **Q7: Limited Domain Evaluation.**
>
> Thank you for the valuable suggestion! We fully agree that evaluating our framework in more diverse driving scenarios would further demonstrate its generalizability and effectiveness. Currently, however, publicly available multimodal autonomous driving datasets are limited and BDD-X and DriveLM are the main public multimodal benchmarks providing both high-level and low-level: as for other datasets, nuScenes lacks high-level action or reasoning annotations, while Waymo does not provide detailed image/ video data. Although richer private datasets exist within industry, these typically cannot be publicly shared due to policy constraints. We sincerely hope future availability of diverse multimodal datasets will enable broader evaluation of methods like ours.

---

### Decision · Program_Chairs · 2025-05-01

**Decision:**

Accept (poster)

**Comment:**

This paper proposes and evaluates SafeAuto, a novel framework for autonomous driving that utilizes multimodal LLMs to address the challenges of integrating high-level prediction and low-level control.

The reviewers identify several strengths of the paper: (1) The SafeAuto framework defines a novel integration of three key components - use of the position-dependent-cross-entropy (PDCE) loss function to improve low-level prediction accuracy, use of Markov Logic Networks (MLN) to factor in explicit safety knowledge, and a contrastive multimodal retrieval automated generation (RAG) for properly aligning video, control signals, and state predicates; (2) The overall SafeAuto framework is shown to produce strong performance against state-of-the-art baselines on two popular benchmark data sets (BDD-X and DriveLM); and (3) the presentation describes the details of the SafeAuto method very clearly (even though it takes some effort to follow).

The weaknesses of the paper that were identified include the following: (1) The experiments are restricted to BDD-X/XDriveLM data sets (for reasonable reasons), which limits the approach's empirical comparison to a narrow number of other approaches and impacts the significance and generality of the results accordingly; (2) despite their integration to form an effective prediction pipeline, the novelty of each component seems marginal, which limits the magnitude of the paper's contribution somewhat; (3) All results appear to have been generated using open-loop simulation, which de-emphasizes the ability of the framework to keep pace with execution; and (4) the presentation is a little verbose in some spots and could also benefit from refinement to make it more relatable to readers working in the autonomous driving field.

Overall, there is consensus that the paper makes an interesting contribution and should be accepted. Please address all of the reviewers comments and suggestions in preparing your final version.